# Single molecule turnover of fluorescent ATP by myosin and actomyosin unveil elusive enzymatic mechanisms

Marko Ušaj [1✉], Luisa Moretto[1], Venukumar Vemula[1], Aseem Salhotra [1] & Alf Månsson [1✉]

Benefits of single molecule studies of biomolecules include the need for minimal amounts of material and the potential to reveal phenomena hidden in ensembles. However, results from recent single molecule studies of fluorescent ATP turnover by myosin are difficult to reconcile with ensemble studies. We found that key reasons are complexities due to dye photophysics and fluorescent contaminants. After eliminating these, through surface cleaning and use of triple state quenchers and redox agents, the distributions of ATP binding dwell times on myosin are best described by 2 to 3 exponential processes, with and without actin, and with and without the inhibitor para-aminoblebbistatin. Two processes are attributable to ATP turnover by myosin and actomyosin respectively, whereas the remaining process (rate constant 0.2–0.5 s$^{-1}$) is consistent with non-specific ATP binding to myosin, possibly accelerating ATP transport to the active site. Finally, our study of actin-activated myosin ATP turnover without sliding between actin and myosin reveals heterogeneity in the ATP turnover kinetics consistent with models of isometric contraction.

[1] Department of Chemistry and Biomedical Sciences, Linnaeus University, SE391 82 Kalmar, Sweden. ✉email: marko.usaj@lnu.se; alf.mansson@lnu.se

Extensive miniaturization by single-molecule methods reduces the need of biological material in biochemical assays with economic, ethical, and scientific implications. Such studies also unveil phenomena that are hidden in ensembles. The first fluorescence-based single biomolecule assay[1] used total internal reflection fluorescence (TIRF) microscopy to study the turnover of a fluorescent ATP analog by the mechano-enzyme myosin II. This molecular motor, underlying muscle contraction and cell motility, is an emerging drug target in severe diseases, such as cancer[2,3], heart failure[4,5], and hypertrophic cardiomyopathy[6] making single molecule ATPase assays of growing interest. However, recent observations of additional exponential processes[7–9] and ~5–50-fold faster rate constants in single myosin molecules[7–9] compared to ensemble studies[10,11] are problematic in this context. Such inconsistencies also hamper the potential to derive rich and reliable mechanistic information from single-molecule fluorescence data[12–16]. Notably, the fast processes, found recently[7–9], have a counterpart in early reports from the 1990s. At the time, however, the faster events were attributed to intrinsic characteristics of expressed myosin constructs (i.e., lack of the light chain-binding neck domain)[15] or to assay limitations (i.e., rebinding of ADP or presence of contaminating particles). The fast events were therefore excluded from the analysis as they were not believed to reflect enzymatic mechanisms[13]. In contrast, recent work[7] associated the fast processes with myosin conformers having different catalytic activity. Clearly, both for the general usefulness of single-molecule data in applications, e.g., drug screening and for deriving reliable mechanistic information, it is essential with unequivocal understanding of the fast processes in relation to enzymatic events with $k_{cat}$ of 0.05–0.1 s$^{-1}$ according to solution kinetics[10,11]. Naturally, if the fast processes just reflect artifacts they should be excluded from the analysis[13] if not, they may convey important mechanistic information per se[7].

Here, we first aim to identify the mechanisms underlying the fast events that tend to dominate over, and sometimes (cf. ref. [7]) completely obscure, true catalytic events. Second, on this basis, we aim to provide solid grounds for a reliable single molecule assay. Our results corroborate a hypothesis that the combination of previously observed fast rate processes and obscured enzymatic events in the single-molecule data reflect a combination of complex dye photophysics (i.e., photobleaching and photoblinking)[17,18] and surface contamination with unidentified fluorescent objects (UFOs). However, interestingly, despite optimizations to remove these confounding effects, we still observe one remaining unexplained exponential process in addition to that consistent with the myosin ATP turnover rate in solution[7–9]. Experimental tests and bioinformatics modeling attribute this unexplained process to nonspecific binding of ATP with a potential role in accelerating ATP transport to the active site. In addition to the studies of myosin alone, our assay optimizations allow unique studies of the >10-fold faster actin-activated myosin ATPase, revealing heterogeneity between individual molecules. This heterogeneity is consistent with an existing theoretical framework with strain-dependent detachment rate of myosin from actin.

## Results and discussion

### Fast rate processes in single-molecule studies under in vitro motility assay conditions

We used objective-based TIRF microscopy (Supplementary Fig. S1) of the fast skeletal muscle myosin II motor fragments heavy meromyosin (HMM) and subfragment 1 (S1) adsorbed to silanized glass surfaces (Figs. 1a and 2a, b, and Supplementary Movies 1 and 2). The fluorescent ATP analog Alexa647-MgATP ("Alexa-ATP" below) that we use, works virtually identically as nonfluorescent MgATP ("ATP" below) with respect to ATP turnover kinetics and powering of

myosin-driven actin filament sliding velocity[19]. Furthermore, in contrast to conventionally used Cy3-based ATP analogs[1,7,8,20], turnover kinetics are similar between different Alexa-ATP isomers[19].

Here, we first focus on the turnover of Alexa-ATP by myosin in the absence of actin (upper part of scheme in Fig. 1b). In solution studies[19,21], the rate limiting step of this process, at the temperature used here (23 °C), is the release of inorganic phosphate (Pi), occurring at a rate of ~0.05–0.1 s$^{-1}$. We performed initial TIRF studies in standard degassed in vitro motility assay (IVMA) solution (Supplementary Methods, section 1.2) similar to that used previously (cf. ref. [7]). This solution contains GOC (glucose, glucose oxidase, and catalase), and DTT as oxygen scavenger and reducing agent, respectively. In further agreement with previous work[7], fluorescence dwell time events (Fig. 2c) were collected from surface "hotspots" (at least ten events during 15 min observation) attributed to myosin motor domains. The resulting cumulative dwell time distributions were fitted by double-exponential functions (Fig. 2e, f) with rate constants (2.7 s$^{-1}$; 55 % and ~0.35 s$^{-1}$; 45 %) appreciably faster than $k_{cat}$ in solution (0.05–0.1 s$^{-1}$). Notably, a rate constant similar to $k_{cat}$ did not resurface in triple exponential fits. These findings corroborate the inconsistencies between single molecules and ensemble data previously demonstrated using Cy3-ATP[7] in similar IVMA solution.

### Single molecule assay optimizations

We hypothesized that UFOs on the motor adsorbing surfaces (Supplementary Fig. S2) was one of the artifacts contributing to the dominance of fast exponential processes in the analysis above. Because such UFOs are stationary, they could be mistaken for hotspots attributed to myosin motor domains. Whereas, their origin and exact photophysical properties are complex they were largely eliminated by the combination of extensive surface cleaning[22], and refined selection and processing of the bovine serum albumin (BSA)[23] (Supplementary Fig. S2) used for surface blocking. Such cleaning and refined BSA selection are therefore central in our optimized assay.

A second class of artifacts that we hypothesized as contributing to a dominance of fast exponential processes with obscured catalytic events is complexities due to dye photophysics (e.g., photobleaching and photoblinking). The potential kinetic effect is evident from a simplified theoretical treatment, assuming mono-exponential bleaching and blinking processes. Under such conditions, the decay rate constant for the observed Alexa-ATP on times on myosin would be the sum of $k_{cat}$, and the rate constants for photobleaching and photoblinking. If the values of the latter are similar to, or higher than, $k_{cat}$ a rate constant with value similar to $k_{cat}$ would disappear from the analysis. As a first test of the importance of such complexities, we varied the illumination intensity fourfold (0.6 ± 0.1 (range) or 2.6 ± 0.1 mW laser power at back focal plane of objective) in experiments performed, using different assay conditions (Supplementary Fig. S3). If photoblinking and/or photobleaching are important one would, a priori, expect a reduction in rate and amplitude of the fast processes[24] with reduced intensity, as well as possible reemergence of the slow process corresponding to catalytic events when performing the assay in IVMA solution. Such effects were however, not clearly observed (Supplementary Fig. S3a, b). Neither did altered illumination intensity noticeably modify the observed rate processes under other experimental conditions (Supplementary Fig. S3c–h). These findings suggest, either that photophysics is not of appreciable relevance or that the range of the intensity variation, achievable using our setup, is insufficient to consistently detect the effect. However, it is also relevant to

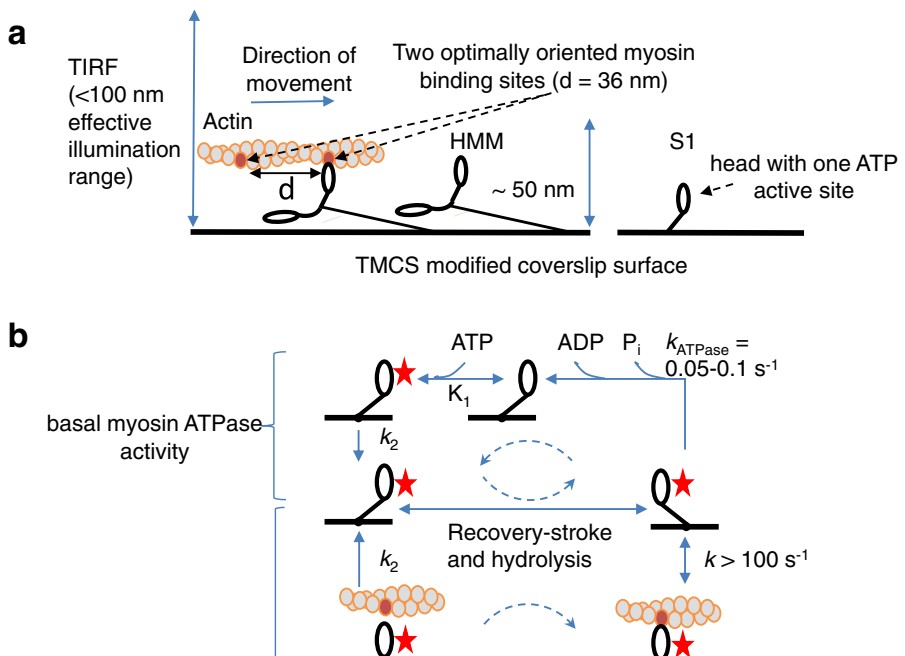

**Fig. 1 Schematics of the study assays and major biochemical transitions. a** TIRF microscopy assay with depicted major proteins used: actin, heavy meromyosin (HMM), and subfragment 1 (S1) attached to TMCS modified coverslip surface. Note that the effective evanescence illumination range is comparable to the distance of myosin catalytic site from the surface. **b** Muscle myosin and actomyosin ATPase cycles. For clarity, only S1 construct is shown. The states with bound nucleotide are marked with a red star. Basal myosin ATPase cycle is rate limited by Pi release, which is significantly accelerated ("activated") by the presence of actin. Dashed circular arrows represent major ATPase cycle direction.

mention that it recently was reported that the origin of Alexa647 dye blinking could be complicated by reversible photoinduced isomerization to at least two dark-state species, in addition to the traditionally accepted role of the triplet excited state[25]. Under certain experimental conditions (i.e., buffer compositions), this could further obscure the blinking process itself and its laser intensity dependence.

To further characterize dye photophysics, isolated from turnover events, we studied Alexa647-phalloidin ("Alexa-Ph") sparsely bound to actin (Supplementary Fig. S4). Instead of varying the laser intensity, we here probed effects of photobleaching and photoblinking by addition of compounds known to[26–29] suppress such events. By sequential addition (for details see Supplementary Information sections 1.2 and 1.9) of redox agents and triple state quenchers (cyclooctatetraene (COT), 4-nitrobenzyl alcohol (NBA), and optimized mixtures of trolox and trolox-hydroquinone (TX/TQ))[18], the rate constant for single-molecule photobleaching/first blinking-event was progressively reduced (Fig. 2d and Supplementary Fig. S4c–g) more than tenfold down to $0.004 \pm 0.00003\,\mathrm{s}^{-1}$ (mean $\pm$ 95% CI). Importantly, Alexa-ATP locked into the nucleotide pocket by vanadate[21,30], exhibited similar photobleaching rate constant as Alexa-Ph, suggesting minor effects of the dye microenvironment. However, the intensity traces (Supplementary Fig. S5) with Alexa-ATP were noisier than with Alexa-Ph. We attribute the

latter effect to thermal fluctuations that cause the myosin motor domains with the active sites to make excursions to different heights above the surface[21], thereby being subjected to varying evanescent wave excitation intensity (cf. Fig. 1a and Supplementary Fig. S5). The excitation intensity decays exponentially away from the surface with a length constant <200 nm, compared to expected height excursions of the myosin motor domain in the range 0–>60 nm (ref. [21]). This idea as partial basis for higher intensity fluctuations of Alexa-ATP than Alexa-Ph-labeled actin filaments is consistent with rather rigid linkage of the actin filaments to the surface via a large number of HMM molecules (in the absence of ATP) per actin filament persistence length—~10 μm (refs. [31,32]). The idea is also consistent with the findings that the intensity variations of Alexa-ATP were reduced by denser surface packing of HMM or increased solution viscosity using methylcellulose (Supplementary Information, sections 1.2, 1.9, and 2.2, and Supplementary Fig. S5), interventions that would inhibit extensive height fluctuations of the motor domains.

To be fully confident that observed hotspots (see above; ref. [7]), after the above optimizations, represent HMM molecules without significant contribution from occasional UFOs, we adsorbed HMM via actin filaments (Fig. 2a) and limited the analysis to positions defined by the filaments (Fig. 2b). Movie 1 illustrates Alexa-ATP dwell time events on HMM molecules that have been deposited on the surface by this approach. Some dwell time

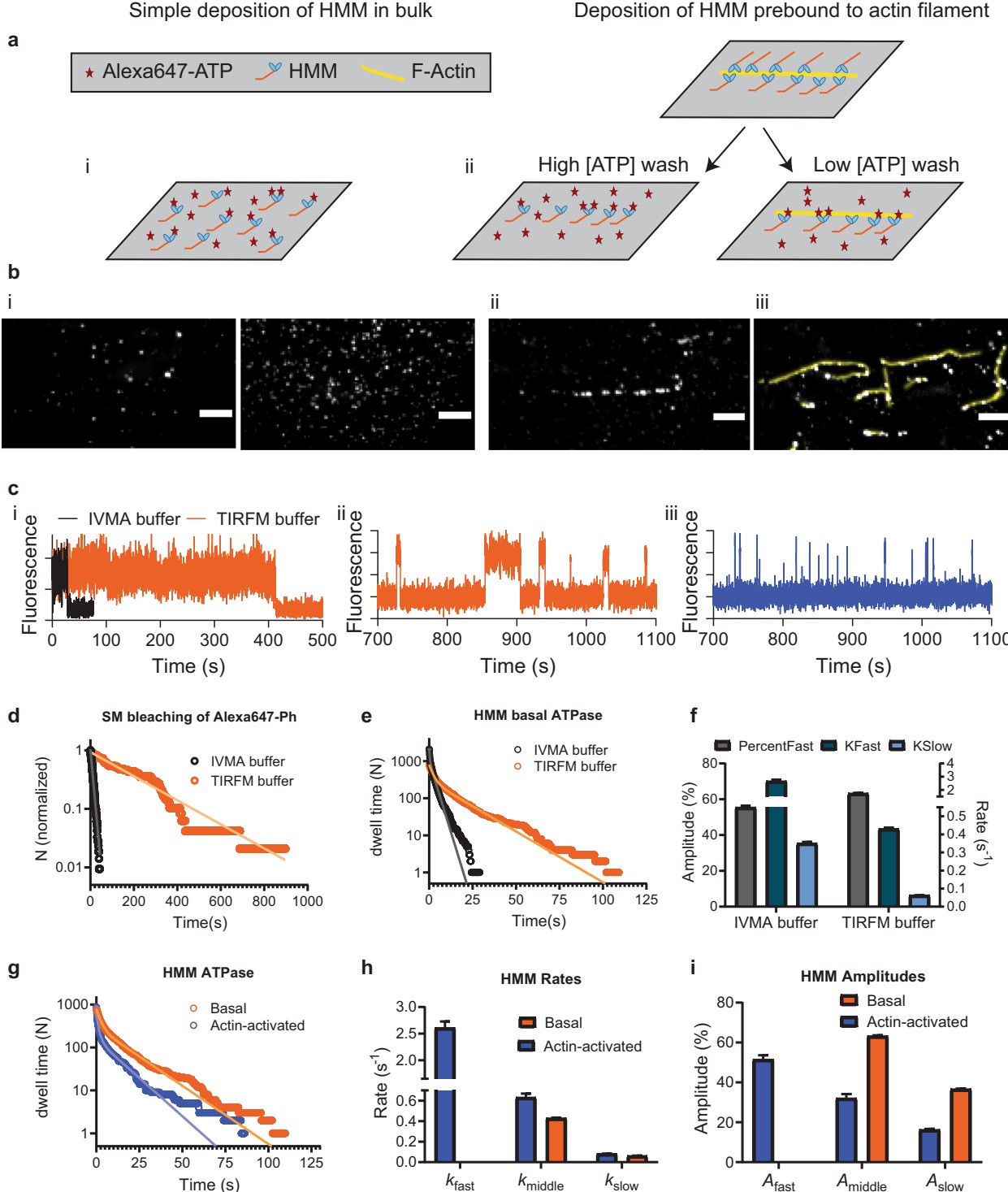

events observed outside the positions defined by the filament represent HMM molecules that have detached from the actin filaments during rinsing, but also occasional remaining UFOs. We took extra care to exclude any remaining effects of the latter type, whether due to nonspecific surface/BSA binding of Alexa-ATP or the Alexa moiety itself (Supplementary Fig. S6).

Because our optimized TIRF assay buffer differs from the standard IVMA assay buffer, we performed IVMA tests in order to examine possible adverse effects of the new additives (Supplementary Information, sections 1.2 and 1.9). The addition of all redox agents and triple state quenchers together had mild

inhibitory effect on actomyosin function (Fig. 3) with a ~25% reduction in actin filament velocity, when the IVMA was performed in TIRF buffer instead of standard IVMA buffer. Reduced velocity (Fig. 3a) can be attributed to the effect of DMSO (2%), used as a vehicle for COT and NBA, and known as reversible inhibitor of actomyosin function[33–37]. This idea is supported by findings that use of TX/TQ alone did not affect velocity (Fig. 3b). However, TX/TQ alone had insufficient suppressing effect on photophysical artifacts: (1) only minor reduction in photobleaching/blinking rate of Alexa647 (Supplementary Fig. S4), (2) only slightly (~50 %) reduced amplitudes

**Fig. 2 Single molecule ATPase. a** Two principles for surface immobilization of myosin motor fragments. **b** Time-averaged Alexa-ATP fluorescence projections of 15 min videos (50 ms exposure time/frame). (i) Simple HMM deposition at 34.3 pM (left) and 343 pM (right). (ii) Optimized HMM deposition following high [ATP] wash. Only spots defined by previous position of F-actin filament were included in analysis. (iii) Image of optimized actomyosin deposition followed by low [ATP] wash showing co-localization of Alexa-ATP fluorescence (gray spots) and F-actin filaments (yellow). Only spots co-localizing with F-actin filaments were included in analysis. Bars, 5 μm. **c** Representative time traces of (i) single-molecule Alexa-Ph bleaching, (ii) HMM basal ATPase, and iii, actin-activated HMM ATPase. Bleaching was observed either using standard in vitro motility assay (IVMA) buffer or optimized TIRF microscopy buffer (TIRFM) as described in the text (for details see Supplementary Information sections 1.2 and 1.9), whereas ATPase traces were from experiments using TIRFM buffer. **d** Cumulative frequency distributions of single-molecule Alexa-Ph until bleach or first blinking event. The distributions are fitted by single exponential functions (solid lines). **e** Cumulative frequency distribution of Alexa-nucleotide dwell time events on HMM surface hotspots (IVMA buffer—simple deposition of HMM, TIRFM buffer-optimized deposition of HMM) were fitted with double-exponential functions (solid lines). IVMA data from 118 HMM molecules, $N_{dwell} = 2112$, TIRFM data from 45 HMM molecules, $N_{dwell} = 785$. **f** Amplitudes and rate constants from the fitting of data in **e**. Note appreciably lower rate constant values in TIRFM compared to IVMA buffer. **g** Cumulative frequency distributions of Alexa-nucleotide dwell time events comparing HMM basal (re-plotted from panel **e**) and actin-activated ATPase activity. Actin-activated ATPase data were obtained from 37 actomyosin hotspots, $N_{dwell} = 879$. The actin–myosin data were fitted with triple exponential functions, whereas basal ATPase data were fitted by double-exponential function as in **f**. **h** Rate constants obtained from fittings to data in **f**. **i** Amplitudes from fitting the data in **f**. Error estimates refer to 95% confidence intervals derived in the regression analysis. Temperature: 23 °C.

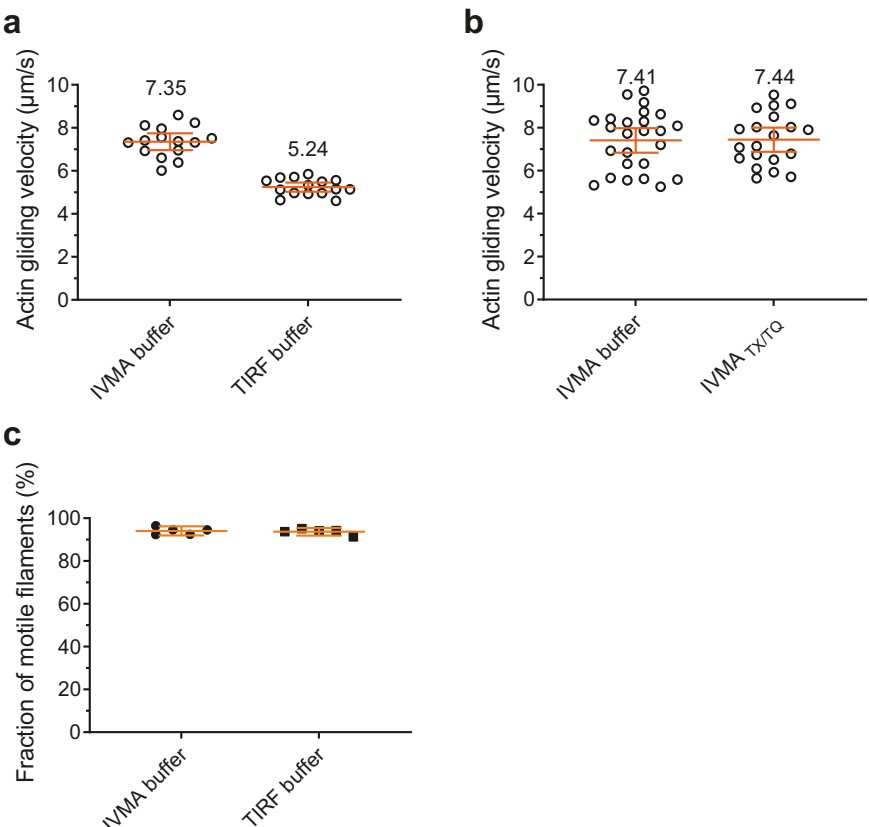

**Fig. 3 Analysis of motility in standard IVMA buffer and optimized TIRF buffer. a** Actin gliding velocity produced by HMM adsorbed to TMCS derivatized glass surfaces. In each individual experiment, 15 actin filaments were analyzed. **b** Actin gliding velocity produced by HMM adsorbed to TMCS derivatized glass surfaces. In each individual experiment, 25 (IVMA buffer) and 20 (IVMA buffer with TX/TQ) actin filaments were analyzed. **c** Fraction of motile filaments from the experiments under **a**, analyzed from five recordings of, in total, 1004 filaments (IVMA buffer) and 925 filaments (TIRF buffer), respectively. Data are given as mean ± 95% confidence interval. Temperature 26 ± 2 °C. For details about buffer composition please see Supplementary Information sections 1.2 and 1.9).

and rates of the fastest (>1 s⁻¹) exponential process associated with Alexa-ATP turnover by myosin (Supplementary Fig. S3b, d), and (3) quite low fractional amplitude (<20%; Supplementary Fig. S3d) of the slow phase reflecting $k_{cat}$. Therefore, we argue that both TX/TQ, COT, and NBA should be included in optimized single-molecule studies of myosin and actomyosin ATPase using Alexa-ATP. This conclusion is supported by findings that: (1) the fraction of motile filaments in the IVMA was not affected, even if both COT, NBA, and TX/TQ were included (Fig. 3c), (2)

reasonable rates were observed for both myosin and actomyosin ATPase[19,21] in the presence of all compounds (see below), and (3) reasonable effects of myosin inhibitors (vanadate and para-aminoblebbistatin; refs. [38,39]) were observed (see below).

**Single molecule myosin basal and actin-activated ATPase under optimized conditions.** Even after the methodological optimizations, dwell time distributions using myosin motor fragments without actin (Fig. 2e, f and Supplementary Fig. S7)

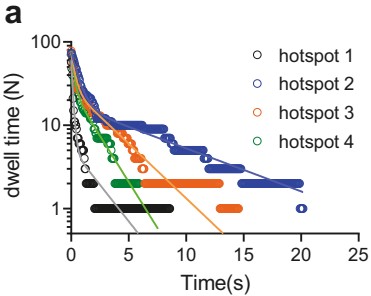
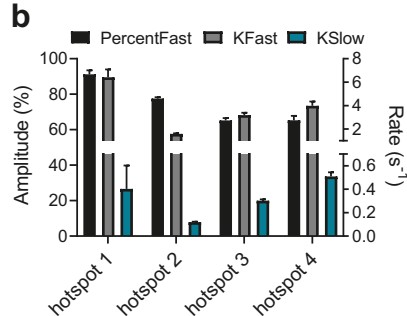

**Fig. 4 Cumulative dwell time distributions recorded from different individual actomyosin hotspots to illustrate the heterogeneity in time constants. a** Cumulative frequency distributions of Alexa-nucleotide dwell time events comparing actin-activated ATPase activity from four different hotspots. Data are fitted with double-exponential functions. **b** Amplitudes and rate constants obtained from fittings to data in **a**. Note that from the individual actomyosin hotspots only two processes attributed to actin-activated ATPase and "unexplained" phase can usually be resolved, while basal myosin ATPase remains obscured. For hotspot 2, it seems likely that the binding site of the HMM molecule is unfavorable for actin attachment, explaining a high ratio of basal to actomyosin ATPase. Error estimates refer to 95% confidence intervals derived in the regression analysis. Temperature: 20 °C.

were better fitted by a double than a single exponential function. However, the rate constants were reduced almost tenfold compared to the situation without optimizations (Fig. 2e, f). Particularly, the fastest phase (rate constant $>2$ s$^{-1}$), ubiquitous and dominant in previous work ($>70\%$)[7], disappeared (Fig. 2f) and was not reinstated by triple exponential fitting (Supplementary Fig. S8) or other changes in the fitting procedure (Table S1). Instead, a slow phase emerged, consistent with basal ATP turnover[10,19,40]. The latter phase also emerged with addition of TX/TQ alone, whereas the fastest phase was not fully removed in this case (Supplementary Fig. S3c, d). With regard to complete removal of the fastest phase by the further addition of COT and NBA, we attribute it to photophysical effects in view of the effects of these compounds on the sum of first photoblinking and photobleaching rate constant in addition to the effect of TX/TQ (Supplementary Fig. S4). We cannot fully exclude that the reduction in rate or merging of the fast ($>1$ s$^{-1}$) and intermediate ($>0.1$ s$^{-1}$) rate constants, seen in the IVMA solution, into one phase (0.2–0.5 s$^{-1}$; 40–70% amplitude; Fig. 2 and Supplementary Fig. S7) is influenced by effects on myosin of COT, NBA, and/or DMSO. However, it is outside the scope of this study to investigate this possibility in detail. First, it does not influence the choice of the optimized assay conditions and second, it does not change the interpretation of the remaining unexplained phase (0.2–0.5 s$^{-1}$) as considered below, only the exact numerical value of its rate.

To summarize, the most important optimizations of the assay (see further Supplementary Information, section 1.9) from the standard developed previously[7] include careful surface treatment, addition of TX/TQ + COT + NBA + methylcellulose and, ideally, myosin surface deposition via actin.

The described optimizations are critical for reliable studies also of actin-activated myosin ATPase (Figs. 1b and 2b, c, iii, g–I, and Supplementary Movie 2). The presence of actin appreciably accelerates the Pi release rate[41], driving the myosin motor to follow an actin-associated rather than an actin-dissociated kinetic path[42]. Several kinetic steps in the actomyosin ATP turnover (lower part of Fig. 1b) are strain sensitive. Strain develops if the myosin motors are fixed to a filament backbone (as in muscle) or to a surface (e.g., our TIRF assay) because the myosin motor domains must be distorted to different degrees, in order to bind to the nearest available actin site. In contrast, soluble myosin motor fragments that interact with actin in solution do not experience any strain. Under such strain-free conditions, the rate limiting transition for the actin-activated myosin ATP turnover cycle is the actin-binding step[43,44]. With surface

adsorbed motors, on the other hand, as in our present experiments, the strain in the cross-bridge elastic element affects several rate constants in the actin-associated kinetic path (cf. refs. [45–48]; Fig. 1b). This includes that related to the release of ADP (and thus Alexa-ADP) from the active site[49]. The latter transition is very fast ($\sim1000$ s$^{-1}$) at zero and "compressive" strains[50] as in solution and toward the end of cross-bridge attachment during fast actin–myosin sliding. When sliding is prevented by external load on a muscle, isometric contraction ensues and the strain formed upon attachment is not dissipated during the attachment period. Under these conditions, the ADP-release rate (and thereby the cross-bridge detachment rate) is 100–1000-fold slower than at zero strain (ref. [51] and references therein, ref. [52]), thus becoming rate limiting for the ATP turnover cycle. If the ADP-release rate is strongly strain dependent one would expect individual myosin motors to exhibit different off rates (cf. ref. [53]; observed as different Alexa-nucleotide dwell time distributions) due to different distortion required to reach their nearest binding site on actin. The ATP (and thus Alexa-ATP) binding to myosin is governed by a second-order rate constant $K_1k_2$ that is generally believed to be of similar magnitude in the presence and absence of actin[41,54].

The very low Alexa-ATP concentration ($\leq10$ nM) that we use in our TIRF experiments, gives "isometric" conditions with undetectable sliding of actin vs myosin. The triple exponential dwell time distribution (Fig. 2g–i and Supplementary Fig. S9) for actin-activated ATP turnover exhibited a fast rate constant of $\sim3$ s$^{-1}$ (51–68%), an intermediate rate constant of 0.5–0.6 s$^{-1}$ (27–32%) and a slow rate constant of 0.05–0.08 s$^{-1}$ (4–16%). Here, we attribute the fastest and slowest phase to actin-activated and basal myosin ATP turnover, respectively, whereas the intermediate phase corresponds to the unexplained phase mentioned above. Notably, the actin-activated ATP turnover rate is slower than 10–17 s$^{-1}$ for isometrically contracting muscle fibers (ref. [51] and references therein), but quite similar to the rate (4 s$^{-1}$) of the slow phase of isometric relaxation upon removal of calcium from a muscle cell[52]. Because detachment from actin of high-force myosin heads dominates the latter process, it is reasonable to presume that our single-molecule data primarily report turnover of high-force heads. A possible basis for faster average "isometric" ATP turnover rate in muscle may be that all sarcomeres are not really isometric, but some shorten against weaker, elongating, sarcomeres[55]. However, strikingly, also under our more strictly controlled isometric conditions, we detect differences in kinetics between individual myosin molecules (Fig. 4). To the best of our knowledge, this is the first direct

observation of such heterogeneity in enzymatic activity. The effect is predicted, by mechanokinetic models of muscle contraction as briefly explained above (e.g., reviewed in ref. [53]). However, we cannot exclude that other mechanisms (e.g., different posttranslational modifications of different enzymes) contribute to the heterogeneity because appropriate control experiments that eliminate strain effects, require appreciably faster time resolution than achievable using our setup (due to 100–1000-fold faster ATP turnover). Also of note is that from the individual actomyosin hotspots (Fig. 4), due to relatively lower number of total events, only two processes—actin-activated ATPase and the "unexplained" phase—can usually be resolved while basal ATPase remains obscured. Thus, the data for individual molecules in Fig. 4 were best fitted by double-exponential functions in contrast to pooled data from many individual molecules, which were consistently best fitted by triple exponentials resolving also remaining basal myosin ATPase activity. Events associated with such activity are severely underrepresented in a single trace from an individual hotspot with a single motor.

Our extension of the studies to actomyosin also revealed reduced average waiting times between dwell time events for actomyosin compared to myosin (Fig. 2c) at 10 nM Alexa-ATP. These results suggest that the second-order rate constant ($K_1k_2$) for Alexa-ATP binding to myosin increases from $(2.80 \pm 0.02)$ $10^6\,M^{-1}\,s^{-1}$ (similar to solution data[50,54]) in the absence of actin to $(4.74 \pm 0.01)$ $10^6\,M^{-1}\,s^{-1}$ in its presence. This issue is further discussed below. Importantly, our methodological optimizations are critical preconditions for faithful analysis of the actomyosin ATP turnover because the fast exponential processes previously found with myosin alone would have seriously disturbed the analysis.

**Origin of unexplained phase in the Alexa-ATP dwell time distributions.** Before going in depth into a mechanistic basis for the unexplained Alexa-ATP-binding phase, it is appropriate to consider possible contribution from remaining trivial complicating factors. Three such factors that seem to be excluded by our control experiments (Supplementary Figs. S2, S6 and S10) are (1) nonspecific binding of the Alexa moiety of Alexa-ATP to myosin/surfaces (Supplementary Fig. S6), (2) Alexa-ADP binding to the active site (Supplementary Fig. S10), and (3) remaining surface binding of Alexa-ATP (Supplementary Figs. S2 and S7). Furthermore, complicating effects due to two myosin heads in HMM are excluded by the similarity between dwell time distributions with S1 (with only one head) and HMM (Figs. 2 and 5, and Supplementary Figs. S7 and S8).

Having thus made trivial complications unlikely as basis for the double-exponential dwell time distributions with myosin alone (Fig. 2), the possibility exists that the double-exponential properties are indeed due to two myosin conformers[7], but with different kinetic properties than suggested previously[7]. Interestingly, however, our observation of nonspecific binding of Alexa-ATP to BSA (Supplementary Fig. S2; see also refs. [56,57]) suggest an alternative origin of the unexplained phase. We thus hypothesize that it reflects nonspecific binding of Alexa-ATP to myosin, outside the active site. Such an effect may effectively keep the ATP molecules in a "cloud" around the myosin surface for a sufficiently prolonged period to infer a binding event with off-rate in the range 0.2–0.5 s$^{-1}$. In order to test this idea, we first blocked the myosin active site by nonfluorescent ATP in the presence of vanadate[38] before adding Alexa-ATP (Fig. 5a–c). If the existence of the unexplained phase reflects a dynamic interconversion between different myosin conformers, i.e., myosin conformations with different catalytic activity upon substrate binding to the active site, it seems inevitable with a reduced amplitude of the

unexplained phase after active site blocking. On the other hand, no such reduction is, a priori expected, if the unexplained phase reflects nonspecific binding outside the active site. In agreement with the latter prediction, active site blocking did not reduce the amplitude of the 0.2–0.5 s$^{-1}$ phase (Fig. 5a–c) despite substantial reduction in amplitude of the slower phase, attributed to basal ATP turnover. In contrast, there was an increase in the absolute number of events per myosin molecule and time (legend, Fig. 5) attributable to the unexplained phase. In addition, a new faster phase became detectable by the fitting algorithm. We postulate that the increase in the total number of unexplained events per molecule and time in these experiments reflects different nonspecific ATP-binding properties in different myosin states.

The results from the experiments with ADP locked to the active site in Fig. 5 were corroborated in experiments (Supplementary Figs. S11 and S12), where fluorescent Alexa-ADP (instead of ADP) was locked into the active site of myosin S1 in the presence of vanadate. Upon subsequent addition of Alexa-ATP, fluorescence intensity events, due to Alexa-ATP binding, were superimposed on top of those attributed to the Alexa-ADP locked by vanadate to the active site (Supplementary Fig. S11). The analysis of these events showed closely similar distributions for the Alexa-ATP-binding events (Supplementary Fig. S12) as when nonfluorescent ADP was locked to the active site (Fig. 5).

These studies, using fluorescent Alexa-ADP locked to the active site, also virtually prove that the unexplained events are not attributed to aborted attempts of Alexa-ATP to bind to active sites of myosin with inaccessible motor domains, e.g., due to (temporary) interactions with the underlying surface. This is in agreement with previous TIRF ensemble studies[19] presenting evidence for additional very slow (but not fast) phases of ATP turnover with amplitude increasing on surface chemistries, where HMM is adsorbed to an increasing degree via its actin-binding, instead of C-terminal tail domain.

The idea of Alexa-ATP-binding sites on HMM outside the active site was corroborated in biochemical experiments using equilibrium dialysis (Supplementary Information sections 1.11 and 2.7, and Supplementary Figs. S13 and S14). These experiments suggested up to ~4 unspecific binding sites per myosin head with binding to such sites at Alexa-ATP concentrations <10 μM (Supplementary Fig. S14c). The findings accord reasonably well with the TIRF data, also quantitatively. Thus, comparison of the Alexa-ATP TIRF data to estimated ATP-affinity to the myosin active site (50–100 nM; inferred from myosin ATPase data in solution[58]), suggests that Alexa-ATP binding, associated with the unexplained process, corresponds to a myosin-Alexa-ATP association constant of 0.2–1 μM. This follows if one assumes: (1) similar association rate constant for Alexa-ATP binding to the active site and to the nonspecific sites that have the highest affinity and (2) four to tenfold faster off-rate constant for the unexplained phase (from the TIRF rates) compared to $k_{cat}$. In the presence of vanadate, as in the equilibrium dialysis experiments, the addition of a process in the TIRF data with up to tenfold higher rates suggests association constant >1 μM, consistent with significant unspecific binding at <10 μM Alexa-ATP in the equilibrium dialysis experiments.

To further investigate the mechanism of nonspecific Alexa-ATP binding to myosin, we performed TIRF experiments at ionic strengths varying from 20 to 130 mM. There was no clear trend for the amplitude and rate constant of the unexplained phase to change within this range of ionic strengths (Supplementary Fig. S15), consistent with the idea that both ionic and nonionic interactions are involved in the nonspecific binding events probed at 10 nM Alexa-ATP. This follows because higher ionic strength contributes to stronger hydrophobic interactions, but weaker electrostatic interactions[59,60].

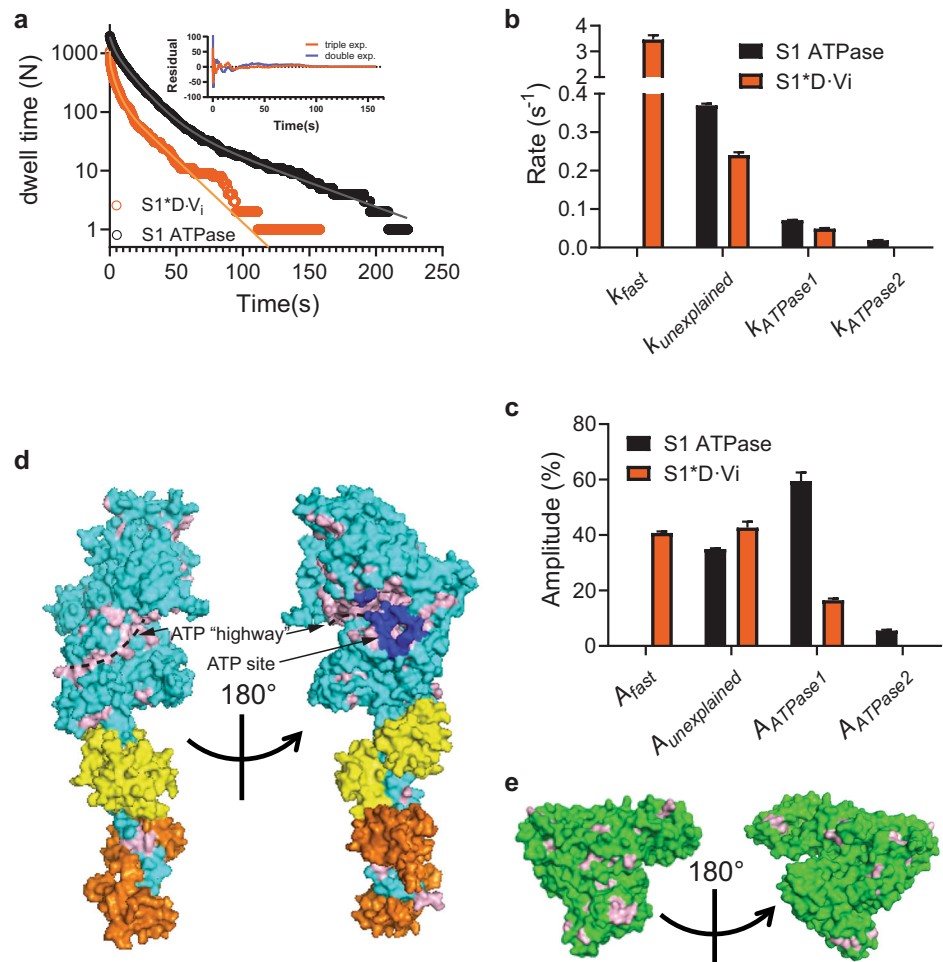

**Fig. 5 Origin of "unexplained" (0.2–0.5 s$^{-1}$) exponential phase. a** Cumulative frequency distributions of Alexa-nucleotide dwell time events on myosin subfragment 1 (S1) surface hotspots (simple deposition of S1, Fig. 1a, i). Data either with ($N_{dwell} = 1964$ from 127 S1 molecules in 15 min) or without ($N_{dwell} = 1084$ events from 30 molecules in 30 min) nucleotide pocket blocking by nonfluorescent ATP and vanadate (S1 * $D \cdot V_i$). The data were fitted by triple exponential functions (solid lines) instead of double-exponential function as justified by residual plots (inset; S1 * $D \cdot V_i$ data), difference in Akaike's information criterion—AICc (4290) and $R^2$ (0.9907 vs 0.9977). **b** Rate constants obtained from fittings to data in **a**. **c** Amplitudes obtained from fitting the data in **a**. Note that amplitude ($A_{ATPase1}$) of basal ATPase activity ($k_{ATPase1}$ ~0.05 s$^{-1}$) was significantly reduced for S1 * $D \cdot V_i$. Unexplained phase with amplitude $A_{unexplained}$ and rate $k_{unexplained}$, was not appreciably affected. Note, further that fitting of S1 * $D \cdot V_i$ data set produced an even faster phase ($k_{fast}$ ~3.5 s$^{-1}$). The extra slow phase with amplitude $A_{ATPase2}$ ~5 % and rate $k_{ATPase2}$ ~0.02 s$^{-1}$ for the S1 ATPase may be attributed to reduced basal S1 ATPase due to S1 attachment with the head to the surface[19]. Error estimates refer to 95% confidence intervals obtained in the fits. Temperature: 23 °C. **d** Model of S1 with predicted surface availability for ATP binding as calculated using software ATPint (see text). Total predicted surface availability for ATP binding (23.7%, pink) vs active site (blue, 1.2%). Proposed "ATP highway" illustrated with dashed line. **e** Model of BSA with predicted surface availability for ATP binding as calculated using ATPint. Total predicted surface availability for ATP binding (13%, pink). Note that unspecific ATP binding to BSA has been experimentally observed (see text).

Whereas we have studied Alexa-ATP rather than ATP, our results also support the idea of nonspecific binding of non-labeled ATP outside the active site of myosin. This is suggested by appreciably higher affinity of Alexa-ATP to myosin outside the active site than of Alexa-ADP (Supplementary Fig. S10c, d) and Alexa-cadaverine (Supplementary Fig. S6) both of which exhibit faster off-rate than Alexa-ATP. However, the affinities of myosin for ATP are probably appreciably lower than for Alexa-ATP, consistent with more potential myosin-binding sites on the rather large Alexa-ATP molecule. Quantitative information would thus be expected to be more challenging to obtain for ATP.

The hypothesis of nonspecific binding, also of nonfluorescent ATP, outside the myosin active site is consistent with experimental results by others[61–63] which however, suggest fewer sites than our equilibrium dialysis experiments with

Alexa-ATP. Nonspecific ATP binding to several sites outside the active site is also supported by results using the bioinformatics tool ATPint[64] (Fig. 5d, e), in contrast to other similar available tools (Supplementary Table S2). Remarkably, after careful tuning of its parameter threshold (Supplementary Fig. S16), the tool predicts substantial ATP interactions on the myosin head surface outside the active site (Fig. 5d)[11], attributing only 8% of the ATP-binding interface to the active site itself. A possible physiological role is that the nonspecific ATP binding provides "highways" funnelling ATP to the active site by faster one-(two-)dimensional compared to three-dimensional diffusion, reminiscent of certain forms of channelling of reaction intermediates over enzyme surfaces in coupled enzymatic reactions[65]. This idea accords with the nonspecific binding sites, according to ATPint, surrounding, and seemingly

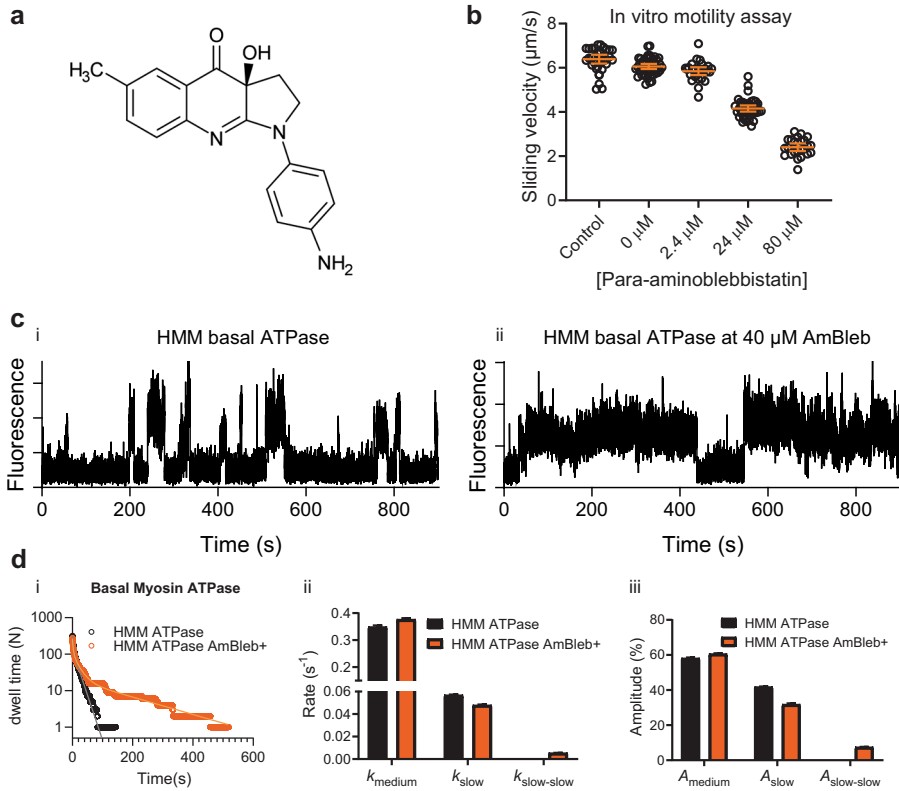

**Fig. 6 Effect of AmBleb on single molecule ATPase. a** The molecular structure of para-aminoblebbistatin (AmBleb) is shown[39]. **b** Concentration–response curve for effect of AmBleb on sliding velocity at 60 mM ionic strength (1 mM MgATP) is shown using HMM. **c** Representative time traces of single molecule basal HMM ATPase without (i) or with 40 μM AmBleb (ii). Note two long dwell time events as direct consequence of effect of AmBleb on basal ATPase activity. **d** Cumulative dwell time distributions (i) of basal HMM ATPase in the absence and presence of 40 μM AmBleb were best fitted to double (control, data from 19 HMM molecules, $N_{dwell} = 318$) or triple (AmBleb, data from 31 HMM molecules, $N_{dwell} = 280$) exponential functions revealing an additional slow phase (ii, iii) with rate constant $0.006 \, s^{-1}$ (~8%) in the presence of AmBleb. Error estimates refer to 95% confidence intervals derived in the regression analysis. Temperature: 20 °C.

radiating out from the active site (Fig. 5d). A similar mechanism has also recently been proposed for the DNA motor RecBCD for which rapid ATP binding outside of the active side is believed to be of functional importance[66]. One may wonder whether accelerated ATP binding would be of importance for actomyosin motor function at intracellular resting concentrations of MgATP which are as high as 5–10 mM (refs. [67,68]). However, first one must keep in mind that both an absolute (due to reduced ATP concentration[67]) and to some extent (after creatine phosphate depletion), relative ATP shortage (due to competition with accumulated ADP) occurs during prolonged or extreme muscle activity. Then, from a quantitative perspective, a dissociation rate constant of actomyosin at 5 mM ATP (including our proposed accelerating mechanism) is ~24,000 $s^{-1}$ (ref. [50]). Without our proposed acceleration by nonspecific ATP binding, the rate constant would be lower and the associated average waiting time until detachment would be >40 μs. Unfortunately, we could not find experimental data for maximal shortening velocities of muscles at physiological temperatures. However, at 30 °C, the observed range of maximum sliding velocities between myosin and actin is 13,000–18,000 nm $s^{-1}$ for mice, rats and rabbits (see ref. [69] and references therein). With a $Q_{10}$ of 2.2 (ref. [50], this would correspond to up to 40,000 nm $s^{-1}$ at physiological body temperatures of 40 °C for rabbits giving a sliding distance, while waiting for ATP induced detachment of up to ~40,000 × $(40 \times 10^{-6})$ nm = 1.6 nm. Such a distance would be associated with an appreciable braking force of the still attached cross-

bridges, strongly supporting the importance of a mechanism to accelerate the ATP binding.

Above, we hypothesized that the degree and kinetics of nonspecific ATP binding may vary between myosin states. If the binding accelerates ATP transport to the active site, it would be physiologically most important in the nucleotide free actin-bound, "rigor" state. This idea fits with our observation of more frequent Alexa-ATP binding to actomyosin than to myosin. Under this interpretation, a fraction of the observed fast binding events with actin present (Fig. 2c, iii) might represent nonspecific ATP binding without actual active site access.

**Effect of para-aminoblebbistatin (AmBleb) on myosin basal ATPase examined on single-molecule level.** The optimized single molecule ATPase assay enabled us to directly observe effects of the myosin inhibitor para-aminoblebbistatin, a light insensitive, nonfluorescent and highly, soluble blebbistatin analog (Fig. 6a)[39,70]. In a proof of principle drug testing experiment (Fig. 6b), we used 40 μM AmBleb that produced half-maximal inhibition of the IVMA sliding velocity, to study ATP turnover by HMM. These studies reveal appreciably longer dwell times in the presence of drug (Fig. 6c, d). For that reason, we could not use the criteria for "hotspots" described above and we analyzed the datasets without this criterion. This shows that certain changes in experimental conditions due to drugs or the presence of mutations call for new and preferably more general criteria for hotspots. Cumulative dwell time distributions of basal ATPase

activity were best fitted using two exponential (control) and three exponential functions (AmBleb) revealing an additional slow phase of $0.005 \, s^{-1}$ in the presence of AmBleb (Fig. 6d). The value of the observed additional slow rate (i.e., $0.005 \, s^{-1}$) is mostly likely photobleaching limited; thus, it represents an upper estimate of the true AmBleb inhibited ATPase activity. The other notable observation is that the rate and amplitude of the "unexplained" ("medium" in Fig. 6) process was little affected by AmBleb despite appreciable effects on the catalytic activity. The latter finding accords with the idea of nonspecific binding of MgATP outside the active site

In conclusion, we report vital optimizations for increased reliability and reproducibility of fluorescence-based single molecule assays. This allows us to unveil previously hidden mechanistic details of the myosin and actomyosin ATPase such as evidence for accelerated ATP transport to the active site and heterogeneities in the actomyosin ATP turnover rate between molecules. These new insights, using 1–100 isolated enzyme molecules (Figs. 2g–i and 4–6), illustrate the generally improved capabilities of the assay. The improvements will benefit a growing spectrum of single-molecule fluorescence-based techniques for studies of actin and myosin, e.g., single-molecule FRET[71], fluorescence polarization[72], and use of nanofabricated surfaces (e.g., zero-mode waveguides) for enhanced fluorescence background suppression[73,74]. The methodological developments are also of value for the very active field of single-molecule fluorescence studies beyond myosin and molecular motors[75]. Finally, the work paves the way for reliable single molecule analysis in high-throughput screening of drug candidates (cf. Fig. 6) with minimal requirement of proteins from costly expression systems, individual cells or clinical samples.

## Materials and methods

**Proteins**. Rabbits for myosin and actin preparations were kept and sacrificed according to procedures approved by the Regional Ethical Committee for Animal experiments in Linköping, Sweden, reference number 73-14. Actin and myosin were prepared from fast skeletal muscle of New Zealand white rabbits (female, 2 kg, 8–9 months old) immediately after sacrifice. Actin was prepared as described earlier[76,77]. Myosin, HMM, and papain S1 were prepared following published protocols (ref. [78], with modifications in refs. [76,79]). Protein preparations were characterized by sodium dodecyl sulfate–polyacrylamide gel electrophoresis, with respect to purity[44] and concentrations were determined spectrophotometrically.

**TIRF microscopy**. We used an objective-type TIRF microscope for all single-molecule experiments. The temperature was 20–25 °C but constant to within 1 °C during a given experiment; see figure legends. The ionic strength was 60 mM unless otherwise stated. For details, see Supplementary Methods, sections 1.3–1.9.

**Dwell time assays**. To follow binding and dissociation of nucleotide, we used Alexa647-ATP as the substrate for myosin. For detailed description of procedures for the dwell time assays, composition of assay buffer, selection criteria for signals of individual molecules, and dwell time analysis, please see Supplementary Methods section 1.10.

**Statistics and reproducibility**. Description of statistical analyses of the data, including central tendency (e.g., means) or other basic estimates (e.g., regression coefficient), variation (e.g., standard deviation), or associated estimates of uncertainty (e.g., confidence intervals) given in figure legends. Besides fitting the sum of exponential functions to the data no other statistical analysis were applicable. We did not perform statistical hypothesis tests, but consider nonoverlapping 95% confidence intervals to indicate statistically significant differences.

Based on the previous work, each binding event of a fluorescent ATP molecule to a myosin molecule was assumed to be an independent random event even on a given experimental occasion using one microscope chamber. The duration of these events were then used to produce cumulative probability distributions as in ref. [7] that were analyzed to give the quantitative data based on exponential fits to the distributions. The number of events used for construction of the distributions ranged between ~100 and 1000 with limited effects of the sample size on the quantitative characteristics of the distributions. This, as well as the reproducibility of the results, is clear from numerous distributions of this type from different experiments (even from individual single molecules) given both in the main paper

(e.g., Figs. 2 and 4) and the Supplementary information (e.g., Supplementary Figs. S7–S9 and S15) supporting our assumption that all individual events are independent. Sample size (i.e., the number of events per cumulative distribution) is readily seen from the graphs (vertical axis) or is denoted in figure legends. No sample size calculation was performed prior to the experiments. Instead we relied on previous information[7,80].

Usually, each cumulative distribution of dwell time events contains events collected from hotspots from one single video per experimental chamber where the field of view was chosen randomly. Videos recorded from different chambers (on the same or different days) served as replicates. Sometimes it was necessary that events from several videos were pooled together to construct the cumulative dwell time distribution. Such pooling was mostly done when using optimized deposition of myosin via actin filaments because the number of hotspots per single video in such experiments was quite limited. The experiments required appreciable specialized skills and were thus performed by a single person who also collected and analyzed the data. However, parts of the IVMAs and single-molecule experiments were collected by one experimenter and analyzed by another person. Some data were also analyzed in duplicate by different persons with negligible differences in results.

**Reporting summary**. Further information on research design is available in the Nature Research Reporting Summary linked to this article.

## Data availability
The datasets generated and/or analyzed during the current study are available from the corresponding author on reasonable request. The Source data underlying Fig. 2c–g are provided as Supplementary Data 2. The Source data underlying Fig. 3a–c are provided as Supplementary Data 3. The Source data underlying Fig. 4a are provided as Supplementary Data 4. The Source data underlying Fig. 5a are provided as Supplementary Data 5. The source data underlying Fig. 6b–d are provided as Supplementary Data 6.

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

## Acknowledgements

This work was funded by European Union Horizon2020 FET Program under contract 732482 (Bio4comp). Further, funding is acknowledged from The Swedish Research Council (grant # 2015-05290) and The Faculty of Health and Life Sciences at The Linnaeus University. Drs. A. Malnasi-Csizmadia, S. Tågerud, M. Norrby, and M.A. Rahman are acknowledged for valuable discussions and suggestions.

## Author contributions

A.M. and M.U. conceived the project, and M.U. and A.M. designed the TIRF system. M.U. and A.M. designed the experiments. M.U. built the TIRF system and performed single-molecule fluorescence experiments. M.U., L.M., and A.M. analyzed TIRF data, and L.M. performed the bioinformatics modeling. V.V. performed in vitro motility assay experiments with and without AmBleb and analyzed the data, and A.S. analyzed in vitro motility assay data, A.M., M.U., L.M., V.V., and A.S. wrote the manuscript and approved the final version.

## Funding

## Competing interests

The authors declare no competing interests.
