## [Peer Review File · Communications Biology]

Reviewers' comments:

Reviewer #1 (Remarks to the Author):

In this work, the authors addressed the kinetics of ATP turnover by myosin from single molecule fluorescence measurements. Through careful optimization of surface treatment and reagent selection as well as rigorous assessment of various components of kinetic rates, they were able to distinguish between different classes of fluorescence burst events originating from fluorescent background, dye photobleaching or blinking, nonspecific binding of ATP, and enzymatic turnover. Dye photophysics and background fluorescence signals are common problems in fluorescence-based single molecule assays that are often overlooked during analysis. Their work bears relevance to virtually every single molecule study that tries to deduce precise kinetics information from a mixture of fluorescence signals. This justifies its appearance in *Communications Biology*, which guarantees visibility to a broad range of readers with potential interest in this work. Their observation of a non-enzymatic kinetic rate which they attributed to the non-specific binding of ATP, with a suggested role in accelerating ATP transport, as well as other findings, is expected to have a large impact in the field of molecular motors. I recommend to publish this work, after addressing the following issues.

1. The current title appears a bit too general, almost sounding like the title of a news article introducing the work. I would like to suggest replacing it with one describing the system of study or the major finding more specifically.

2. When there exists a mixture of kinetics from dye photophysics (bleaching and blinking) and actual enzymatic reactions, the most straightforward way to distinguish them would be to change the illumination intensity and check if each kinetic rate or its relative population changes. Of course, the intensity should be as high as to reliably distinguish the signal from background fluctuation and as low as not to let photobleaching overwhelm the whole kinetics. But I believe there still is a room to tune the intensity at least 3-5 fold. With increasing illumination strength, I would expect the kinetics originating from blinking to get faster and higher in relative population, while the kinetic rates and relative populations of the fast and slow enzymatic reactions should not change, ideally. The effects of dye-stabilizing reagents like TX/TQ, COT, and NBA have been carefully examined but this kind of experiment would provide a direct evidence that the eliminated kinetics is purely photophysical rather than any side effect of those chemicals on the activity of myosin.

3. Molecule-to-molecule heterogeneity was reported for actin-bound myosin and it was attributed to the varying surface attachment of myosin resulting in varying strain in individual actomyosin complex. This might be physiologically relevant considering the varying local environment of myosin monomers. As it would be straightforward to prove this by showing that a free standing myosin does not bear such heterogeneity, I wonder if it is possible to find an assay to observe myosin sitting on surface-bound actin filaments or suspended actin filaments.

4. Non-specific binding of ATP was found to have a rate of 0.2-0.5 s⁻¹. As it is a relatively slow process for a non-specific binding event, I wonder if such stable non-specific binding of ATP can be justified from former reports or their own computational prediction.

5. It is a bit difficult to grab the overall picture of the kinetics revealed in this study from quick reading. I would like to recommend to add a schematic diagram as a main figure, showing the diverse branches of kinetics (both enzymatic and non-productive) and denoting their rates, relative populations, and dependence on the inherent conformation or external environment.

Thank you.

Sincerely,

Hajin Kim, Ph.D.

Reviewer #2 (Remarks to the Author):

The authors have fluorescence based TIRF based single molecule microspectroscopic tools to unveil the mechanistic details about the ATP turnover rates by myosin and actomyosin to provide a glimpse of enzymatic reaction pathways that are not possible to get from ensemble studies. It is well known that mechano enzyme Myosin II utilizes the process of ATP hydrolysis to carry out motility functions such as contraction of muscles, however from single molecule studies it was clear that this rate of ATP turnover by myosin occurs having two different reaction kinetics, some of the myosin molecules showing ATP turnover at a faster rate while some myosin showing it with a slower kinetic rate. The analysis and the smFRET Experiments by M. Amrute-Nayak suggests that the probable reason could be due to interconversion between two different Myosin conformers. It is been observed that the rate of ATP turnover estimated by single molecule measurements are of approximately 5-50 times faster compared to the ensemble measurements. However, a suitable explanation for this behaviour is somehow missing in the discussion section, though the possibilities of interference of dye photophysics and hence the measure of optimizations to remove these phenomena has been considered.

The triple exponential dwell time distribution by adsorbing myosin to actin filaments, it was observed that actin activated ATP turnover showed a fast rate phase(2-3 s⁻¹), an intermediate rate phase(0.4-0.5 s⁻¹) and a slow rate phase(0.05-0.08 s⁻¹). The fast and slow rate kinetics unambiguously refers to the ATP turnover by actin activated and basal myosin molecules. They have tried to hypothesize occurrence of the intermediate i.e the "unexplained" phase due to the non-specific binding of ATP to different myosin conformers state outside the active site. In presence of myosin inhibitor para-aminoblebbistatin, the "unexplained" process is little affected although the catalytic effect is adversely affected, also supports their hypothesis of non specific ATP binding to myosin surface as the reason behind the generation of phase with intermediate rate constant. It would have been nice if the authors could identify the unexplained state.

The authors have done careful experiments and their conclusions are supported by the data. However, there are issues as explained above and some specific issues as stated below to be addressed before consideration for publication. Many of the places, there is lack of proper explanation of the observed results. Some of the points as mentioned below

1. The introduction part appeared to be missing out on certain points as follows, and including them would make the paper to be transmitted on to wider scientific domain. The topic could have been introduced in a way that would inform the readers a little in actin myosin dynamics during muscle contraction so that they would get an exposure to the background before being taken to the technical matters. It could also state the hypothesis in a clear way while defining the topic.
2. It would be nice to give a proper explanation with proper citing, why thermal fluctuation predominates Alexa-ATP compared to Alexa Ph.

Reviewer #3 (Remarks to the Author):

In this study Usaj and co-workers refine a single molecule imaging approach to reveal a new binding site on myosin for ATP. Using TIRF microscopy and HMM or S1 deposited onto the surface randomly or with actin to guide the delivery the authors measure the attached lifetimes of fluorescent ATP. To achieve better quality imaging the investigators used a number of blocking/cleaning strategies, which had the effect of increasing the attached lifetimes of the same dye but attached to phalloidin.

Overall the study addresses an important area and the experiments contain a number of reasonable controls. The data quality was good and statistically believable. However, the major concern for this reviewer was the interpretation of the results. I believe the authors also struggled over this and the final explanation for the 'odd' rate constant was the result of elimination of other

possibilities. I was also confused over the use of HMM and S1 in the study, it would have been cleaner to include just S1 in the main study and HMM in the supplemental; however, I suspect that S1 was not compatible with the IVMA and actin deposition experiments.

Major concerns:

1. What stops the heads from landing on the surface in any orientation? They could land motor domain down and even if they don't initially, nothing stops them from transiently interacting with the surface. In such a situation, is it possible that the $0.5s^{-1}$ rate is due to surface-induced effects? The problem is that in all situations from S1 to HMM to amleb, this rate does not change and its amplitude remains constant, implying it is not an active process – but it could be a surface effect as much as a myosin surface effect.
2. The authors do not use an orthogonal assay to confirm the $0.5s^{-1}$ rate is due to binding to the myosin surface.
 - a. There is no evidence in the field that 2 alexa-ATP molecules bind to a single myosin head. Do the authors ever see two overlapping fluorescent spots indicative of two ATPs bound?
 - b. Based on the lifetime and their own calculation of the 2nd order alexa-ATP binding I estimate an affinity of $0.2 \mu M$. To be certain that they have two molecules bound a biochemical estimation of the stoichiometry of alexa-ATP binding would be essential.
 - c. Is there a salt dependence to the binding lifetime of alexa-ATP?
3. Can the authors fully exclude photobleaching from the slow rate? Did they perform these lifetime determinations at a few laser powers?
4. I am not convinced by the ATPint software. While assessing this study I tried this software on another ATPase. It detects Walker motifs well, but then it seems to assign so many residues to ATP binding that one is able to create any story to match the output of this program; it is concerning that only 8% of matched residues lay in the canonical ATP binding pocket. In addition, the authors did not reveal the detection threshold.

Minor:

Line 62: should read - that "are" hidden

Throughout: More specific reference to the buffers and conditions that are present in the supplemental. I would suggest a more organised supplemental with a contents page and section headings that are referenced from the main text.

Line 156: Fig 2b is the wrong figure reference

Fig 3: Why were these only fit to doubles and not triples as they should be?

Line 200: Fig 3c is not there and the labelling of b is oddly placed. In fact this references fig 4 anyway!

Line 216: Figs S6-7 show only the basal ATPase activities, but what happens with D.Vi? Does the fast phase emerge, what about the actin activated ATPase of S1, what does that look like?

Line 227: the braking force argument is predicated on an in vivo concentration of $1mM$ ATP, however this could be up to $10mM$ so the braking force experienced would be much slower. This also questions the use of a second site in vivo.

Line 247: Amrute-Nayak and co-workers in their 2014 paper used an antibody to lift their protein from the surface, this study might benefit from a similar strategy.

Line 252: There is no Fig 5e

Dear Reviewers,

We are grateful for a careful and constructive review of our manuscript that has helped us to achieve major improvements. Please find below our detailed point to point response to the reviewer comments with the original comments in italics. In addition, the key changes are also indicated by highlighted text in a submitted version of the manuscript for review only. Please note, that a list of references that we cite below are located after the responses to both the editor and all reviewers.

Response to Reviewer #1

1. The current title appears a bit too general, almost sounding like the title of a news article introducing the work. I would like to suggest replacing it with one describing the system of study or the major finding more specifically.

Response and changes made: We have changed the title along the suggested lines.

2. When there exists a mixture of kinetics from dye photophysics (bleaching and blinking) and actual enzymatic reactions, the most straightforward way to distinguish them would be to change the illumination intensity and check if each kinetic rate or its relative population changes. Of course, the intensity should be as high as to reliably distinguish the signal from background fluctuation and as low as not to let photobleaching overwhelm the whole kinetics. But I believe there still is a room to tune the intensity at least 3-5 fold. With increasing illumination strength, I would expect the kinetics originating from blinking to get faster and higher in relative population, while the kinetic rates and relative populations of the fast and slow enzymatic reactions should not change, ideally. The effects of dye-stabilizing reagents like TX/TQ, COT, and NBA have been carefully examined but this kind of experiment would provide a direct evidence that the eliminated kinetics is purely photophysical rather than any side effect of those chemicals on the activity of myosin.

Response: We acknowledge the importance of control experiments of the suggested type. However, results in the literature⁶ imply that effects of altered illumination intensity on blinking time and blinking fraction may, to some degree, vary between fluorophores and experimental conditions, despite the fact that it is usually assumed that higher laser power yields higher blinking fraction (e.g. ⁷). Experimental results at varied illumination intensity need to be viewed against this background. Nevertheless, during our methods developments (buffer compositions, myosin depositions, and coverslip cleaning method) we have frequently performed assays at two different laser powers of 0.6 and 2.6 mW (as measured at the back focal plane); technical limitations prevented us from extending this range. Generally, our results showed no consistent effects of varied illumination intensity on the rate constants and fractional amplitudes of different exponential phases in the Alexa-ATP on-dwell-time distributions. This is now illustrated in new supporting data (new Fig. S3) for different conditions. We further agree with the suggestion by the reviewer that one cannot, a

priori, exclude effects on myosin as basis for parts of the effects on the Alexa-ATP dwell time distributions seen when introducing all of the compounds TX/TQ, COT and NBA. We have therefore performed new experiments and analyses as well as expanded the discussion to more fully address this issue. Together, with the studies at different illumination intensities, the results (new and from previous version) may be summarized as follows: 1 (new). The photophysics of the Alexa647 fluorophore does not change appreciably with changes in illumination intensity in the range tested (Fig. S3), 2 (previous). TX/TQ alone has, compared to TX/TQ+COT+NBA, only moderate reducing effect on the sum of the photobleaching and photobleaching rate constant of Alexa647 (Fig. S4d), 3 (previous). A slow phase in the dwell-time distributions ($0.05\text{-}0.1\text{ s}^{-1}$) that is attributed to basal myosin ATPase is observed with addition of either TX/TQ (giving a slow phase of low amplitude) or TX/TQ, COT and NBA (giving a slow phase of appreciably higher fractional amplitude) (Fig. 4), 4 (new). TX/TQ alone reduces the amplitude and rate of the fastest ($>1\text{ s}^{-1}$) exponential phase in the dwell time distributions to less degree than introduction of TX/TQ+COT+NBA (Fig. S3), 5 (new/previous). TX/TQ alone does not reduce the HMM propelled actin filament sliding velocity in the in vitro motility assay (Fig. 3b) but TX/TQ+COT+NBA reduces the velocity by $\sim 25\%$ (Fig. 3a), 6 (new/previous). Neither TX/TQ⁸ nor TX/TQ+COT+NBA reduces the fraction of motile filaments in the in vitro motility assay (Fig. 3c). These results (1-6) lead us to conclude that: A. Photophysics effects have an important role in obscuring the slowest phase in the dwell time distributions as the slow ($0.05\text{-}0.1\text{ s}^{-1}$) phase is reintroduced by the addition of TX/TQ alone (that has no effects on actin sliding velocity). B. The fastest phase in dwell-time distributions has a contribution from photophysical effects as the amplitude and rate of this phase is reduced by addition of TX/TQ alone. However, here we do not exclude that other factors contribute as discussed in different places in the manuscript (see below). The observed, rather surprising, behavior of the Alexa647 dye blinking process in relation to laser power can be related to recently reported complexities of the origin of Alexa647 dye blinking, involving reversible photo-induced isomerization to at least two long-lived dark species in addition to traditionally accepted role of the triplet excited state⁹. Under certain experimental conditions (i.e. buffer compositions) this could further obscure the mechanism of blinking process itself and its laser intensity dependence.

Changes made: We have added new data (Fig. S3) to address the issue raised in this point in greater detail. We have also considered the findings in context with previously available data in several places in the main paper: page 4, paragraph 3. Furthermore, the complexities are discussed in further detail in the main paper on page 5, paragraphs 3-4. Finally, based on the new information and discussion we have somewhat weakened the conclusions on page 5, paragraph 3.

3. Molecule-to-molecule heterogeneity was reported for actin-bound myosin and it was attributed to the varying surface attachment of myosin resulting in varying strain in individual actomyosin complex. This might be physiologically relevant considering the varying local environment of myosin

monomers. As it would be straightforward to prove this by showing that a free standing myosin does not bear such heterogeneity, I wonder if it is possible to find an assay to observe myosin sitting on surface-bound actin filaments or suspended actin filaments.

Response: This is an interesting and important suggestion. However, the actomyosin ATP turnover in the absence of strain (free-standing myosin) is too fast ($>50 \text{ s}^{-1}$) to be readily accessible using our camera and illumination system with too few photons captured for the short exposure times needed. An analysis that we could in principle perform is to compare variability of myosin basal ATPase with the variability in the isometric actomyosin ATPase. The hypothesis behind such an approach would be that basal ATPase active would vary less than actomyosin ATPase if variability is strain-related. Whereas we did consider probing and comparing variability of myosin ATP turnover rate in the absence of actin, such experiments also turned out to be prohibitively complex for several reasons. The most important complication is much longer average Alexa-ATP on-dwell-times ($\sim 20 \text{ s}$; due to slower ATP turnover) and lower frequency of events with Alexa-ATP binding to myosin (average waiting time between events $\sim 50 \text{ s}$ at 10 nM Alexa-ATP; see paper). These properties make it virtually impossible to collect sufficient number of events (say $> \sim 30$; ≥ 57 used for actomyosin ATPase in Fig. 4) from individual molecules. This is particularly challenging as we wish to limit our collection period to $< 30 \text{ min}$ to prevent functional deterioration of surface adsorbed myosin molecules. Another limitation of using myosin ATP turnover for control purposes in the probing of heterogeneity between molecules, is that myosin ATPase is rate limited by a different kinetic step than the isometric actomyosin ATP turnover (see new Fig. 1). Finally, studies of the myosin ATP turnover will add complexities due to some HMM molecules being adsorbed to the surface solely via their motor domain giving a 10-fold slower rate constant ($\leq 0.02 \text{ s}^{-1}$; cf. ⁸) than with basal myosin ATPase in solution. The latter complication is not affecting actomyosin ATPase because such HMM molecules would not bind actin.

Changes made: We have now explicitly (on page 6, paragraph 2) discussed the limitations outlined above. Additionally, we have weakened our statements related to strain dependent rates in the abstract and Introduction (final lines) as well as on page 6.

4. Non-specific binding of ATP was found to have a rate of 0.2-0.5 s⁻¹. As it is a relatively slow process for a non-specific binding event, I wonder if such stable non-specific binding of ATP can be justified from former reports or their own computational prediction.

Response: This is an issue that we struggled with quite a lot before initial submission and it was also raised by the other reviewers and highlighted by the editor. We have therefore addressed it in new experiments and analyses as follows:

Most importantly, we performed equilibrium dialysis experiments (Figs. S13-14) suggesting Alexa-ATP binding to up to 4 sites per myosin head outside the active site with association constants $< 10 \text{ } \mu\text{M}$. This is consistent with our TIRF results in the presence of vanadate as further elaborated on in

new text in the paper. Vanadate inhibition of the myosin ATPase was used in the equilibrium dialysis experiments for practical reasons.

Additionally we have justified the bioinformatics evidence for ATP binding outside the active site by further description of the software ATPint used for this purpose, as well as further analyses using this and other freely available software.

Finally, after an extended literature search we found already existing experimental evidence¹⁻³ for more than one ATP binding site on myosin subfragment 1 (and thus HMM) and very recent evidence⁴ for non-specific ATP binding sites outside the active site of the DNA-based motor enzyme RecBCD. Interestingly, the non-specific binding of ATP to RecBCD was innovatively proposed in a recent study⁴ to play similar role as we propose for the non-specific binding of ATP to myosin. In the new version of the manuscript we discuss the data in relation to these previous findings. In summary, we provide both orthogonal experimental evidence (by us and others) and strengthened theoretical arguments for weak ATP binding outside the active site of myosin.

Notably, in our equilibrium dialysis experiments we use Alexa-ATP rather than ATP. This is motivated for practical reasons and on basis of the major aim of the paper to produce an appreciably refined single molecule fluorescence based ATPase assay. We would here like to emphasize the great importance of this improvement in itself. Thus, a previous study (e.g. Amrute-Nayak et al. ⁵) led those authors to present an idea of myosin conformers but without Alexa-ATP fluorescence dwell-time events of a rate ($\sim 0.05 \text{ s}^{-1}$) consistent with the basal myosin ATP turnover in solution ensemble studies. The confusing results and the hypotheses emerging from them (largely falsified by our work) not only lead the myosin field astray. More generally, the use of fluorescence based single molecule enzyme assays, is cast in doubt, independent of enzyme system. In view of these arguments, the behavior of the fluorescent Alexa-ATP rather than ATP itself is of greatest relevance in the present paper. Despite this key goal however, our results also support non-specific binding of non-labelled ATP outside the active site of myosin as mentioned before^{2,3}. This follows from appreciably higher affinity of Alexa-ATP to myosin outside the active site than of Alexa-ADP (higher off-rate than ATP; Fig. S10) and Alexa-cadaverine (lower binding and higher-off rate than ATP; Fig. S6). However, the affinities of myosin for non-specifically bound ATP are probably appreciably lower than for Alexa-ATP, because of more potential myosin binding sites on the rather large Alexa-ATP molecule including addition of additional net charges (-3) due to the Alexa moiety (see Discussion on page 8, paragraph 3). Quantitative information is thus expected to be appreciably more challenging to obtain for ATP. We also believe that such studies are outside the scope of the paper because they would cause the paper to strongly diverge away from its central aim.

In addition to the equilibrium dialysis experiments, we have also performed new TIRF experiments to further characterize the non-specific Alexa-ATP binding. First, we performed single molecule experiments with Alexa-ADP locked to the active site in the presence of vanadate while adding Alexa-ATP

to the solution (new Figs. S11-12). These experiments showed events with short periods of double intensity superimposed on the long Alexa-ADP dwell times in the presence of vanadate. This is direct evidence for simultaneous binding of one Alexa-ATP molecule outside the active site in addition to the Alexa-ADP molecule locked to the active site by vanadate. In view of the large number of binding sites found in the equilibrium dialysis experiments one may wonder if it would not also be possible to observe events representing more than two simultaneously bound Alexa-ATP molecules outside the active site, i.e. with higher intensities than double the intensity of one Alexa-ATP molecule. However, in view of the quite short dwell times the probability is extremely low for more than one simultaneous non-specific binding event in addition to the active site binding at the Alexa-ATP concentrations that we can readily use.

To characterize the non-specific Alexa-ATP binding further we performed experiments at different ionic strengths (Fig. S15). These experiments did not appreciably change the number and rates associated with non-specific binding events suggesting that both ionic and non-ionic interactions are involved in the non-specific Alexa-ATP binding (because higher ionic strength contributes to stronger hydrophobic interactions but weaker electrostatic interactions^{10,11}). This is also in accordance with the ATPint data suggesting that amino acids with both basic (16 %) and hydrophobic (46 %) side-chains are involved in the areas with predicted binding.

Changes made: The new experiments (Equilibrium dialysis, Alexa-ATP dwell times in the presence of Alexa-nucleotide and vanadate, Alexa-ATP dwell times at varied ionic strength) and modelling are described in detail in the Supporting Information (Supporting Materials and Methods, Sections 1.11-1.12; Supporting Results and Discussion text, sections 2.6-2.9 and Figs. S11-16). The key findings from the new experiments are also described in relation to work by others in the main paper on pages 7-8 in several paragraphs.

5. It is a bit difficult to grab the overall picture of the kinetics revealed in this study from quick reading. I would like to recommend to add a schematic diagram as a main figure, showing the diverse branches of kinetics (both enzymatic and non-productive) and denoting their rates, relative populations, and dependence on the inherent conformation or external environment.

Response and changes made: Such a diagram has now been included as a new Figure 1 of the paper.

Response to Reviewer #2:

The authors have fluorescence based TIRF based single molecule microspectroscopic tools to unveil the mechanistic details about the ATP turnover rates by myosin and actomyosin to provide a glimpse of enzymatic reaction pathways that are not possible to get from ensemble studies. It is

well known that mechano enzyme Myosin II utilizes the process of ATP hydrolysis to carry out motility functions such as contraction of muscles, however from single molecule studies it was clear that this rate of ATP turnover by myosin occurs having two different reaction kinetics, some of the myosin molecules showing ATP turnover at a faster rate while some myosin showing it with a slower kinetic rate. The analysis and the smFRET Experiments by M. Amrute-Nayak suggests that the probable reason could be due to interconversion between two different Myosin conformers. It is been observed that the rate of ATP turnover estimated by single molecule measurements are of approximately 5-50 times faster compared to the ensemble measurements. However, a suitable explanation for this behaviour is somehow missing in the discussion section, though the possibilities of interference of dye photophysics and hence the measure of optimizations to remove these phenomena has been considered.

The triple exponential dwell time distribution by adsorbing myosin to actin filaments, it was observed that actin activated ATP turnover showed a fast rate phase(2-3 s⁻¹), an intermediate rate phase(0.4-0.5 s⁻¹) and a slow rate phase(0.05-0.08 s⁻¹). The fast and slow rate kinetics unambiguously refers to the ATP turnover by actin activated and basal myosin molecules. They have tried to hypothesize occurrence of the intermediate i.e the "unexplained" phase due to the non-specific binding of ATP to different myosin conformers state outside the active site. In presence of myosin inhibitor para-aminobebistatin, the "unexplained" process is little affected although the catalytic effect is adversely affected, also supports their hypothesis of non specific ATP binding to myosin surface as the reason behind the generation of phase with intermediate rate constant. It would have been nice if the authors could identify the unexplained state.

Response: We agree that this issue is highly important. We struggled with it quite a lot before initial submission and it was also raised by the other reviewers and highlighted by the editor. We have therefore addressed it in new experiments and analyses as follows. Most importantly, we performed equilibrium dialysis experiments (Fig. S13-14) suggesting Alexa-ATP binding to up to 4 sites per myosin head, outside the active site, with association constants <10 μ M. This is consistent with our TIRF results in the presence of vanadate as further elaborated on in new text in the paper (see "Changes made" below). Vanadate inhibition of the myosin ATPase was used in the equilibrium dialysis experiments for practical reasons. Additionally we have better justified the bioinformatics evidence for ATP binding outside the active site by further description of the software ATPint used for this purpose, as well as further analyses using this and other freely available software.

Finally, an extended literature search revealed already existing experimental evidence¹⁻³ for more than one ATP binding site on myosin subfragment 1 (and thus HMM) and very recent evidence⁴ for non-specific ATP binding sites outside the active site of the DNA-based motor enzyme RecBCD. Interestingly, the non-specific binding of ATP to RecBCD was innovatively proposed in the recent study⁴ to play similar role as we propose for the non-specific binding of ATP to myosin. In the new version of the manuscript we discuss the data in relation to these previous findings. In summary we provide both orthogonal experimental evidence (by us and others) and

strengthened theoretical arguments for weak ATP binding outside the active site of myosin.

Notably, in our equilibrium dialysis experiments, we use Alexa-ATP rather than ATP. This is motivated for practical reasons and on basis of the major aims of the paper to produce an appreciably refined single molecule fluorescence based ATPase assay. We would here like to emphasize the great importance of this improvement in itself. Thus, a previous study (e.g. Amrute-Nayak et al. ⁵) led those authors to present an idea of myosin conformers but without Alexa-ATP fluorescence dwell-time events of a rate ($\sim 0.05 \text{ s}^{-1}$) consistent with the basal myosin ATP turnover in solution ensemble studies. The confusing results and the hypotheses emerging from them (largely falsified by our work) may not only lead the myosin field astray. More generally, the use of fluorescence based single molecule enzyme assays is cast in doubt, independent of enzyme system. In view of these arguments, the behavior of the fluorescent Alexa-ATP rather than ATP itself is of greatest relevance in the present paper. Despite this key goal however, our results also support non-specific binding of non-labelled ATP outside the active site of myosin as mentioned before^{2,3}. This follows from appreciably higher affinity of Alexa-ATP to myosin outside the active site than of Alexa-ADP (higher off-rate than ATP; Fig. S10) and Alexa-cadaverine (lower binding and higher-off rate than ATP; Fig. S6). However, the affinities of myosin for non-specifically bound ATP are probably appreciably lower than for Alexa-ATP, because of more potential myosin binding sites on the rather large Alexa-ATP molecule including addition of additional net charges (-3) due to the Alexa moiety⁶ (see Discussion on page 8, paragraph 3). Quantitative information would thus be appreciably more challenging to obtain for ATP. We believe that such studies are outside the scope of the paper because they would cause the paper to strongly diverge away from its central aim.

In addition to the equilibrium dialysis experiments, we have also performed new TIRF experiments to further characterize the non-specific Alexa-ATP binding. First, we performed experiments with Alexa-ADP locked to the active site in the presence of vanadate while adding Alexa-ATP to the solution (new Fig. S11-12). These experiments showed events with short periods of double intensity superimposed on the long Alexa-ADP dwell times in the presence of vanadate as direct evidence for simultaneous binding of one Alexa-ATP molecule outside the active site in addition to the Alexa-ADP molecule locked to the active site. In view of the large number of binding sites found in the equilibrium dialysis experiments one may wonder if it would not also be possible to observe events representing more than two simultaneously bound Alexa-ATP molecules outside the active site, i.e. with higher intensities than double the intensity of one Alexa-ATP molecule. However, in view of the quite short dwell times (related to low affinity) the probability is extremely low for more than one simultaneous non-specific binding event in addition to the active site binding at the Alexa-ATP concentrations that we can readily use.

To characterize the non-specific Alexa-ATP binding further we performed experiments at different ionic strengths (Fig. S15). These experiments

showed that altered ionic strength did not appreciably change the number and rates associated with non-specific binding events suggesting that both ionic and non-ionic interactions are involved in the non-specific Alexa-ATP binding (because higher ionic strength contributes to stronger hydrophobic interactions but weaker electrostatic interactions^{10,11}). This is in accordance with the ATPint data suggesting that amino acids with both basic (16 %) and hydrophobic (46 %) side-chains are involved in the areas with predicted binding.

Changes made: The new experiments (Equilibrium dialysis, Alexa-ATP dwell times in the presence of Alexa-nucleotide and vanadate, Alexa-ATP dwell times at varied ionic strength) and modelling are described in detail in the Supporting Information (Supporting Materials and Methods, Sections 1.11-1.12; Supporting Results and Discussion text, sections 2.6-2.9 and Figs. S11-16). The key findings from the new experiments are also described in relation to work by others in the main paper on pages 7-8 in several paragraphs.

The authors have done careful experiments and their conclusions are supported by the data. However, there are issues as explained above and some specific issues as stated below to be addressed before consideration for publication. Many of the places, there is lack of proper explanation of the observed results. Some of the points as mentioned below

Response: We have now scanned the paper for parts with lack of proper explanation for the results throughout and have specifically addressed the points raised below.

Changes made: Proper explanations were added to the observed results in several parts of the manuscript, e.g. page 4 (paragraph 2-4), page 5 (paragraph 3-4), etc.

1. The introduction part appeared to be missing out on certain points as follows, and including them would make the paper to be transmitted on to wider scientific domain. The topic could have been introduced in a way that would inform the readers a little in actin myosin dynamics during muscle contraction so that they would get an exposure to the background before being taken to the technical matters. It could also state the hypothesis in a clear way while defining the topic.

Response and changes made: The Introduction is supported by new parts of the Results and Discussion section (last paragraph on page 3, last paragraph on page 5 – first paragraph on page 6) accompanied by a new Fig. 1 to introduce myosin and actomyosin dynamics in greater detail.

With regard to hypothesis statements, we have now added new explicitly stated hypotheses in addition to the key hypothesis stated in the second paragraph of the discussion on the basis for fast exponential processes obscuring catalytic events. The new explicitly stated hypotheses have been added in the second and third paragraphs of page 4 and early in the third paragraph of page 7.

2. *It would be nice to give a proper explanation with proper citing, why thermal fluctuation predominates Alexa-ATP compared to Alexa Ph.*

Response: This issue has now been considered with proper citations. Thermal fluctuations of the myosin motor domains of HMM cause them (with their active sites) to make excursions to different heights above the surface¹² thereby being subjected to varying evanescent wave excitation intensity. The excitation intensity decays exponentially away from the surface with a length constant around 100 nm, to be compared to the expected height excursions of the myosin motor domain in the range 0 - >60 nm¹². This idea as basis for higher intensity fluctuations of Alexa-ATP than Alexa-Ph labelled actin filaments is consistent with rather rigid linkage of the actin filaments to the surface via a large number of HMM molecules (in the absence of ATP) per actin filament persistence length - approximately 10 μm ¹³. The idea is also consistent with the findings that the intensity variations of Alexa-ATP were reduced by denser packing of HMM on the surface or increased solution viscosity using methylcellulose.

Changes made: The explanation given above has now been added to the main text on page 4, final paragraph – page 5, first paragraph. Proper citations have also been added.

Reviewer #3 (Remarks to the Author):

In this study Usaj and co-workers refine a single molecule imaging approach to reveal a new binding site on myosin for ATP. Using TIRF microscopy and HMM or S1 deposited onto the surface randomly or with actin to guide the delivery the authors measure the attached lifetimes of fluorescent ATP. To achieve better quality imaging the investigators used a number of blocking/cleaning strategies, which had the effect of increasing the attached lifetimes of the same dye but attached to phalloidin.

Overall the study addresses an important area and the experiments contain a number of reasonable controls. The data quality was good and statistically believable. However, the major concern for this reviewer was the interpretation of the results. I believe the authors also struggled over this and the final explanation for the 'odd' rate constant was the result of elimination of other possibilities. I was also confused over the use of HMM and S1 in the study, it would have been cleaner to include just S1 in the main study and HMM in the supplemental; however, I suspect that S1 was not compatible with the IVMA and actin deposition experiments.

Response: We agree with the reviewer that S1 for many reasons would be “cleaner” to use. However, the use of S1 also comes with complexities. One of these is that suggested by the reviewer, that S1 did not work well with the actin deposition experiments. Second, the use of S1 would be more amenable to surface effects as the head is necessarily always very close to the surface (cf.¹²). Nevertheless, for certain purposes (e.g. one head vs two head per molecule), it is important to also show and analyze results for S1 single molecule behavior motivating that we do not only describe HMM effects in the main paper but also S1 data.

Major concerns:

1. What stops the heads from landing on the surface in any orientation? They could land motor domain down and even if they don't initially, nothing stops them from transiently interacting with the surface. In such a situation, is it possible that the 0.5s-1 rate is due to surface-induced effects? The problem is that in all situations from S1 to HMM to ambleb, this rate does not change and its amplitude remains constant, implying it is not an active process – but it could be a surface effect as much as a myosin surface effect.

Response: We agree that the myosin heads do land on the surface in different orientations as described in our previous detailed studies^{8,12,14} although surface immobilization via the head is limited for HMM on TMCS derivatized surfaces^{8,12,14}. In these earlier TIRF ensemble studies we found evidence for an additional *very slow (not fast!) phase* of ATP turnover with amplitude varying between surface chemistries. Whereas it might be that we have missed a fast phase in the previous work due to limited time resolution, the evidence from those studies provide one piece of argument against the idea that fast processes appear with surface binding of the head. Additional, seemingly stronger evidence is: 1. the similar behavior of surface-adsorbed HMM and S1, 2. the observation of non-specific binding events in equilibrium dialysis experiments (with HMM in solution) and, finally, 3. our newly reported findings (Fig. S11-12) that the fast events occur on top of events where Alexa-ADP is locked into the active site of myosin S1 by vanadate. The latter result virtually proves that the unexplained events are not attributed to e.g. aborted attempts of Alexa-ATP to bind to the active site of a (temporarily) surface adsorbed motor domain of myosin.

Changes made: The possibility of surface effects is raised on page 7, (second last paragraph from the bottom) in connection with the new Alexa-ATP/Alexa-ADP experiments also providing evidence against one specific type of surface effect. The other changes, e.g. related to equilibrium dialysis and locking of Alexa-ADP to myosin by vanadate are considered in greater detail below.

2. The authors do not use an orthogonal assay to confirm the 0.5s-1 rate is due to binding to the myosin surface.

a. There is no evidence in the field that 2 alexa-ATP molecules bind to a single myosin head. Do the authors ever see two overlapping fluorescent spots indicative of two ATPs bound?

b. Based on the lifetime and their own calculation of the 2nd order alexa-ATP binding I estimate an affinity of 0.2 μ M. To be certain that they have two molecules bound a biochemical estimation of the stoichiometry of alexa-ATP binding would be essential.

c. Is there a salt dependence to the binding lifetime of alexa-ATP?

Response to 2a-2c: We agree that these issues (2a-c) are of great interest and, just as assumed by the reviewer, we struggled with them quite a lot before initial submission. Similar issues were also raised by the other reviewers and highlighted by the editor. We have therefore addressed them in depth in new experiments and analyses as follows:

Most importantly, we performed equilibrium dialysis experiments (Figs. S13-14) suggesting Alexa-ATP binding to up to 4 sites per myosin head

outside the active site with association constants $< 10 \mu\text{M}$. This is consistent with our TIRF results in the presence of vanadate as further elaborated on in new text in the paper (see “Changes made”, below). Vanadate inhibition of the myosin ATPase was used in the equilibrium dialysis experiments for practical reasons.

Additionally we have better justified the bioinformatics evidence for ATP binding outside the active site by further description of the software ATPint used for this purpose, as well as further analyses using this and other freely available software (Fig. S16, Table S2).

Finally, an expanded literature search revealed experimental evidence¹⁻³ in the literature for more than one ATP binding site on myosin subfragment 1 (and thus HMM) and very recent evidence⁴ for non-specific ATP binding sites outside the active site of the DNA-based motor enzyme RecBCD.

Interestingly, the non-specific binding of ATP to RecBCD was originally proposed in a recent study⁴ to play similar role as we propose for the non-specific binding of ATP to myosin. In the new version of the manuscript we discuss the data in relation to the mentioned results in the literature. In summary, we provide both orthogonal experimental proofs (by us and others) and strengthened theoretical arguments for weak ATP binding outside the active site of myosin.

Notably, in our equilibrium dialysis experiments, we use Alexa-ATP rather than ATP. This is motivated for practical reasons and on basis of the major aims of the paper to produce an appreciably refined single molecule fluorescence based ATPase assay. We would here like to emphasize the great importance in itself of the latter improvement. Thus, a previous study (i.e. ⁵) led those authors to present an idea of new myosin conformers but without Alexa-ATP fluorescence dwell-time events of a rate ($\sim 0.05 \text{ s}^{-1}$) consistent with the basal myosin ATP turnover in solution ensemble studies. The confusing results and the hypotheses emerging from them (largely falsified by our work) may not only lead the myosin field astray. More generally, the use of fluorescence based single molecule enzyme assays is cast in doubt, independent of enzyme system. In view of these arguments, the behavior of the fluorescent Alexa-ATP rather than ATP itself is of greatest relevance in the present paper. Despite this key goal, however, our results also support non-specific binding of non-labelled ATP outside the active site of myosin as mentioned before^{2,3}. This follows from appreciably higher affinity of Alexa-ATP to myosin outside the active site than of Alexa-ADP (higher off-rate than ATP; Fig. S10) and Alexa-cadaverine (lower binding and higher-off rate than ATP; Fig. S6). However, the affinities of myosin for non-specifically bound ATP are probably appreciably lower than for Alexa-ATP, consistent with more potential myosin binding sites on the rather large Alexa-ATP molecule also with addition of net charges (-3) due to the Alexa moiety⁶. Quantitative information would thus be appreciably more challenging to obtain for ATP. We further believe that such studies are outside the scope of the paper because they would cause the paper to strongly diverge away from its central aim.

In addition to the equilibrium dialysis experiments, we have also performed new TIRF experiments to further characterize the non-specific Alexa-ATP

binding. First, we performed experiments with Alexa-ADP locked to the active site in the presence of myosin while adding Alexa-ATP to the solution (new Figs. S11-12). These experiments showed events with short periods of double intensity superimposed on the long Alexa-ADP dwell times in the presence of vanadate as direct evidence for simultaneous binding of one Alexa-ATP molecule outside the active site in addition to the Alexa-ADP molecule locked to the active site. In view of the large number of binding sites found in the equilibrium dialysis experiments one may wonder if it would not also be possible to observe events representing more than two simultaneously bound Alexa-ATP molecules outside the active site, i.e. with higher intensities than double the intensity of one Alexa-ATP molecule. However, in view of the quite short dwell times (related to low affinities), the probability is extremely low for more than one simultaneous non-specific binding event in addition to the active site binding at the Alexa-ATP concentrations that we can readily use.

To characterize the non-specific Alexa-ATP binding further we performed experiments at different ionic strengths (Fig. S15). These experiments showed that altered ionic strength did not appreciably change the number and rates associated with non-specific binding events suggesting that both ionic and non-ionic interactions are involved in the non-specific Alexa-ATP binding (because higher ionic strength contributes to stronger hydrophobic interactions but weaker electrostatic interactions^{10,11}). This is in accordance with the ATPint data suggesting that amino acids with both basic (16 %) and hydrophobic (46 %) side-chains are involved in the areas with predicted binding.

Changes made: The new experiments (Equilibrium dialysis, Alexa-ATP dwell times in the presence of Alexa-nucleotide and vanadate, Alexa-ATP dwell times at varied ionic strength) and modelling are described in detail in the Supporting Information (Supporting Materials and Methods, Sections 1.11-1.12; Supporting Results and Discussion text, sections 2.6-2.9 and Figs. S11-16). The key findings from the new experiments are also described in relation to work by others in the main paper on pages 7-8 in several paragraphs.

3. Can the authors fully exclude photobleaching from the slow rate? Did they perform these lifetime determinations at a few laser powers?

Response and changes made: Previous and new experimental findings suggest that photobleaching has negligible effect on the slow rate (0.05-0.1 s⁻¹) attributed to the basal myosin ATPase. First, this phase showed similar rate (although lower amplitude) in the presence of TX/TQ alone as in the presence of both TX/TQ, COT and NBA (Fig. S4), conditions where photobleaching/photoblinking exhibit 10-fold difference in rate (Fig. 2). Further, new experiments, using 4-fold difference in illumination intensity under different conditions (Fig. S3), showed no difference in the rate of the slow process corresponding to basal myosin ATPase. This is now also mentioned and discussed on pages 4-5 of the main paper.

However, at this point we are somewhat uncertain about exactly which slow rate the reviewer actually refers to. For instance, a rate slower than then basal myosin ATPase in solution is observed in certain basal ATPase experiments and explained as basal ATPase of myosin attached with the motor domain to the surface. This extra slow rate is most likely partly photobleaching limited as rates as low as $\sim 0.001 \text{ s}^{-1}$ have been observed in ensemble studies⁸. Similarly the slowest rate observed as a consequence of myosin ATPase inhibitors (vanadate and AmBleb) used in this study is also, most likely, photobleaching limited. This has now been clarified in main text on page 9, paragraph 2.

4. I am not convinced by the ATPint software. While assessing this study I tried this software on another ATPase. It detects Walker motifs well, but then it seems to assign so many residues to ATP binding that one is able to create any story to match the output of this program; it is concerning that only 8% of matched residues lay in the canonical ATP binding pocket. In addition, the authors did not reveal the detection threshold.

Response and changes made: Our use of the ATPint software has been described in appreciably greater detail in the present version of the paper and we have further considered its validity (Table S2, Fig. S16 and related Supporting text). More about the ATPint software in relation to other new results is also considered under point 2 above.

Line 62: should read - that “are” hidden
Corrected. (3d line of the Introduction).

Throughout: More specific reference to the buffers and conditions that are present in the supplemental. I would suggest a more organised supplemental with a contents page and section headings that are referenced from the main text.

Response and changes made: We agree that this approach appreciably improves the structure of the paper. Therefore, the SI has now been re-organized with a contents page and section headings. However, in order to avoid a range of bewildering cross-references we have limited direct references to text sections in the SI. Instead we have generally referenced the supporting Figures but in each legend of these figures we have given a reference to associated sections in the Supporting text. However, direct references from the main paper to sections in the SI are given in some key instances e.g. in the main Materials and Methods section, the second last paragraph on page 5 and the last paragraph on page 7.

Line 156: Fig 2b is the wrong figure reference

Response and changes made: Corrected. Now “Fig. 3c”, towards end of third paragraph on page 5.

Fig 3: Why were these only fit to doubles and not triples as they should be?

Response: When actomyosin or dwell times of many molecules are pooled together the best fit to the cumulative dwell time distribution is seen with a triple exponential function. However, when individual molecules (hotspots) are probed it happens that (due to relatively lower number of total events)

that certain phases are not resolved well. The major reason in our case that individual actomyosin data are fitted to double exponentials is pretty straightforward: events which present basal ATPase activity are severely underrepresented in one such trace. Only by pooling several myosins (hotspots) together, their number becomes sufficiently large to be captured by the fitting algorithm.

Changes made: The above explanation is briefly repeated in the revised manuscript in the main text on page 6, end of paragraph 2, supplementing brief information previously given in the legend of current Fig. 4 (previous Fig. 3).

Line 200: Fig 3c is not there and the labelling of b is oddly placed. In fact this references fig 4 anyway!

Response and changes made: Corrected. (Now reference to Fig. 5a-c in middle of third paragraph on page 7).

Line 216: Figs S6-7 show only the basal ATPase activities, but what happens with D.Vi? Does the fast phase emerge, what about the actin activated ATPase of S1, what does that look like?

Response: This comment refers to the present Figs. S8-S9. We performed D.Vi experiments only using S1. The reason is that in this specific case, the approach with one, rather than two heads, is important and gives much cleaner results. Thus, with S1, there is only one head per hotspot that needs to be effectively occupied by D.Vi in contrast to HMM where there are two. Thus, experiments using HMM run the risk of giving confusing results depending on whether 0, 1 or 2 heads (all of which may subsequently bind Alexa-ATP) have D.Vi at the active site. In view of these complexities we did not perform any experiments with D.Vi and HMM.

The actin-activated ATPase of S1 was not studied due to the experimental challenges. As already predicted by the reviewer, S1 did not work well with the actin deposition experiments i.e. we could not get S1 deposition via actin filament in an ordered line depicting the actin filament. Rather, S1 was deposited randomly regardless of use of actin. We did not further pursue the reason for that.

Line 227: the braking force argument is predicated on an in vivo concentration of 1mM ATP, however this could be up to 10 mM so the braking force experienced would be much slower. This also questions the use of a second site in vivo.

Response: We apologize for being a bit sloppy in our arguments in this context by mainly considering orders of magnitude. We agree that the MgATP concentrations reported for living muscle cells during rest usually ranges between about 5 and 10 mM^{15 16}. However an absolute (due to reduced ATP concentration¹⁵) and to some extent (after PCr depletion), relative ATP shortage (due to competition with accumulated ADP) may occur due to prolonged or extreme muscle activity. Further, a dissociation rate constant of actomyosin at 5 mM MgATP (including our proposed mechanism) would be approximately 24 000 s⁻¹ (from Nyitrai and Geeves¹⁷). Thus, without the proposed mechanism the rate constant would be lower than

24 000 s⁻¹ and the associated average waiting time until detachment would be >40 μs. Further, we mentioned a sliding velocity of 10 000 nm/s as an order of magnitude estimate. However, the real value is actually appreciably higher. First, at 30 °C the observed range is actually 13 000 -18 000 nm/s (see ¹⁸ and references therein). With a Q₁₀ of 2.2¹⁷, this range would instead be 22000 – 40000 nm/s at physiological body temperatures of 37-40 °C in humans and frequently used experimental animals (e.g. mice, rats, rabbits [used here]). For the highest velocity in this range (rabbit) the sliding distance while waiting for ATP induced detachment would be 40 000 x 40 10⁻⁶ nm = 1.6 nm suggesting that our previously stated arguments are valid.

Changes made: We have now briefly updated the text on page 8 (second last paragraph) with the above information while we maintain that the proposed mechanism may be of value in reducing braking effects during fast shortening.

Line 247: Amrute-Nayak and co-workers in their 2014 paper used an antibody to lift their protein from the surface, this study might benefit from a similar strategy.

Response: Amrute-Nayak and co-workers⁵ used a variety of different myosins. When they used rabbit (psoas, soleus) muscle myosin they deposited it by a similar direct surface adsorption approach as used here, most notably without the use of antibodies. Only when they used recombinant Dictyostelium discoideum myosin-2, myosin-5b and myosin-1B constructs modified by his tags at the N terminal they used the antibody approach. We are also planning to use such an approach for our expressed myosins (work in progress). However, the addition of any new protein component to the assay comes with the risk of adding new complexities e.g. as illustrated by the Alexa-ATP binding to BSA considered in our present paper (cf. particularly Fig. S2). We therefore prefer not to tread the antibody immobilization path at this stage.

Line 252: There is no Fig 5e

Response and changes made: Corrected (towards end of 2nd paragraph on page 9), i.e. this has now been changed to Fig. 6d (corresponding to Fig. 5d in the previous version).

REFERENCES:

- 1 Eccleston, J. F. Fluorescence changes associated with the binding of ribose-5-triphosphate to myosin subfragment 1: Evidence for a second triphosphate binding site. *FEBS Letters* **113**, 55-57, doi:[https://doi.org/10.1016/0014-5793\(80\)80493-5](https://doi.org/10.1016/0014-5793(80)80493-5) (1980).
- 2 MRAKOVČIĆ-ZENIC, A., REISLER, E. & ORIOL-AUDIT, C. On the Alkali Light Chains of Vertebrate Skeletal Myosin. *European Journal of Biochemistry* **115**, 565-570, doi:10.1111/j.1432-1033.1981.tb06240.x (1981).

- 3 Gyimesi, M. *et al.* The mechanism of the reverse recovery step,
phosphate release, and actin activation of Dictyostelium myosin II. *J*
Biol Chem **283**, 8153-8163, doi:10.1074/jbc.M708863200 (2008).
- 4 Zananiri, R. *et al.* Auxiliary ATP binding sites power rapid
unwinding by RecBCD. *bioRxiv*, 210823, doi:10.1101/210823
(2018).
- 5 Amrute-Nayak, M. *et al.* ATP turnover by individual myosin
molecules hints at two conformers of the myosin active site.
Proceedings of the National Academy of Sciences of the United States
of America **111**, 2536-2541, doi:10.1073/pnas.1316390111 (2014).
- 6 Vandenberk, N., Barth, A., Borrenberghs, D., Hofkens, J. & Hendrix,
J. Evaluation of Blue and Far-Red Dye Pairs in Single-Molecule
Förster Resonance Energy Transfer Experiments. *The Journal of*
Physical Chemistry B **122**, 4249-4266, doi:10.1021/acs.jpcc.8b00108
(2018).
- 7 Bartley, L. E., Zhuang, X., Das, R., Chu, S. & Herschlag, D.
Exploration of the Transition State for Tertiary Structure Formation
between an RNA Helix and a Large Structured RNA. *Journal of*
Molecular Biology **328**, 1011-1026,
doi:https://doi.org/10.1016/S0022-2836(03)00272-9 (2003).
- 8 Balaz, M., Sundberg, M., Persson, M., Kvassman, J. & Månsson, A.
Effects of Surface Adsorption on Catalytic Activity of Heavy
Meromyosin Studied using Fluorescent ATP Analogue. *Biochemistry*
46, 7233-7251 (2007).
- 9 Karlsson, J. K. G., Laude, A., Hall, M. J. & Harriman, A. Photo-
isomerization of the Cyanine Dye Alexa-Fluor 647 (AF-647) in the
Context of dSTORM Super-Resolution Microscopy. *Chemistry – A*
European Journal **25**, 14983-14998, doi:10.1002/chem.201904117
(2019).
- 10 Zhang, Z., Yang, Y., Tang, X., Chen, Y. & You, Y. Effects of Ionic
Strength on Chemical Forces and Functional Properties of Heat-
induced Myofibrillar Protein Gel. *Food Science and Technology*
Research **21**, 597-605, doi:10.3136/fstr.21.597 (2015).
- 11 Hu, Y. *et al.* Dye adsorption by resins: Effect of ionic strength on
hydrophobic and electrostatic interactions. *Chemical Engineering*
Journal **228**, 392-397, doi:https://doi.org/10.1016/j.cej.2013.04.116
(2013).
- 12 Persson, M. *et al.* Heavy Meromyosin Molecules Extend more than
50 nm above Adsorbing Electronegative Surfaces. *Langmuir* **26**,
9927-9936 (2010).
- 13 Vikhorev, P. G., Vikhoreva, N. N. & Mansson, A. Bending flexibility
of actin filaments during motor-induced sliding. *Biophys J* **95**, 5809-
5819 (2008).
- 14 Mansson, A. Translational actomyosin research: fundamental insights
and applications hand in hand. *Journal of muscle research and cell*
motility **33**, 219-233, doi:10.1007/s10974-012-9298-5 (2012).
- 15 Kuno, S.-y. & Itai, Y. Muscle Energetics during Exercise by
³¹P NMR. *The Annals of physiological anthropology*
11, 313-318, doi:10.2114/ahs1983.11.313 (1992).

- 16 Kushmerick, M. J., Moerland, T. S. & Wiseman, R. W. Mammalian skeletal muscle fibers distinguished by contents of phosphocreatine, ATP, and Pi. *Proceedings of the National Academy of Sciences* **89**, 7521-7525, doi:10.1073/pnas.89.16.7521 (1992).
- 17 Nyitrai, M. *et al.* What limits the velocity of fast-skeletal muscle contraction in mammals? *J. Mol. Biol.* **355**, 432-442 (2006).
- 18 Mansson, A., Persson, M., Shalabi, N. & Rassier, D. E. Nonlinear Actomyosin Elasticity in Muscle? *Biophys J* **116**, 330-346, doi:10.1016/j.bpj.2018.12.004 (2019).

REVIEWERS' COMMENTS:

Reviewer #1 (Remarks to the Author):

The authors have addressed all the raised concerns successfully. By performing carefully designed additional experiments and rigorously interpreting the results, they unambiguously assigned the observed kinetics to the enzymatic turnovers and non-productive binding of ATP. I am happy to recommend publishing this work in the current form.

Reviewer #2 (Remarks to the Author):

The Authors have taken suitable measure in carrying out the experiments suggested and in data analysis to answer the concerns raised during the previous review. The Modified manuscript is now well written and will be appreciated by a large group of readers. The experiments and the findings are well supported by the figures. Recommended for publication.

Reviewer #3 (Remarks to the Author):

I feel that this revised version of the manuscript is significantly improved as a result of the suggestions from the first review. This study offers a non-canonical view of how myosin acquires its ATP. For me the confirmation of this would come from overlapping fluorophores on a single myosin. They do present some data to support this, and therefore addresses my query. However, I remain skeptical, but I think it important for the myosin community to judge this work because as a reviewer my role is to ensure quality of science (which this meets), not to impede controversial results being published.

One further point I would like to make about ATPint, having tried this software myself on known ATPases I do not feel confident about its use.

Finally, there are a few language lapses that should be considered, ATPase site and incorrectly been dubbed ATPase side in a few locations (notably figure 1) in the manuscript.

Dear Reviewers,

We are grateful for the final comments on our manuscript and the valuable suggestions throughout this review process that have helped us to achieve major improvements. Please find below our detailed point to point response to the final reviewer comments with the original comments in italics.

REVIEWERS' COMMENTS:

Reviewer #1 (Remarks to the Author):

The authors have addressed all the raised concerns successfully. By performing carefully designed additional experiments and rigorously interpreting the results, they unambiguously assigned the observed kinetics to the enzymatic turnovers and non-productive binding of ATP. I am happy to recommend publishing this work in the current form.

Response to Reviewer #1

We are grateful for your final remarks!

Reviewer #2 (Remarks to the Author):

The Authors have taken suitable measure in carrying out the experiments suggested and in data analysis to answer the concerns raised during the previous review. The Modified manuscript is now well written and will be appreciated by a large group of readers. The experiments and the findings are well supported by the figures. Recommended for publication.

Response to Reviewer #2

Thank you for your final remarks!

Reviewer #3 (Remarks to the Author):

I feel that this revised version of the manuscript is significantly improved as a result of the suggestions from the first review. This study offers a non-canonical view of how myosin acquires its ATP. For me the confirmation of this would come from overlapping fluorophores on a single myosin. They do present some data to support this, and therefore addresses my query. However, I remain skeptical, but I think it important for the myosin community to judge this work because as a reviewer my role is to ensure quality of science (which this meets), not to impede controversial results being published.

One further point I would like to make about ATPint, having tried this software myself on known ATPases I do not feel confident about its use.

Finally, there are a few language lapses that should be considered, ATPase site and incorrectly been dubbed ATPase side in a few locations (notably figure 1) in the manuscript.

Response to Reviewer #3

First we would like to thank you for your final remarks. We salute the attitude of being skeptical yet open to new controversial ideas. Having great trust in our data and analysis we are, nevertheless, also looking forward to future independent confirmation(s) or refutation(s) of our interpretation of the results.

We believe we have provided solid evidence for overlapping fluorophores on a single myosin although as always there is room for improvement. Additional pieces of evidence might emerge from the use of super-resolution localization microscopy to pinpoint fluorophore centers by nano or even sub-nanometer accuracy. Unfortunately, however, this technology is currently not available in our lab.

In our hands, ATPint was the only software among many tested which predict unspecific binding. This was an important selection criterion as we view it as likely (considering significant charge and possibilities of hydrophobic interactions) that a range of proteins should show certain degree of unspecific ATP binding, from BSA (probably without functional meaning) to myosin and any other ATPase (with likely functional importance as discussed in our paper). We believe that our changes related to ATPint in the previous revised version (in response to this reviewer), including more detailed description of the program and comparison to other software will allow the readers to more critically judge the results of the ATPint software.

We have taken care of the mentioned language lapses and also carefully scanned the manuscript for other similar errors. Thank you again for careful reading.